# An integrative proteomics method identifies a regulator of translation during stem cell maintenance and differentiation

Pierre Sabatier [1], Christian M. Beusch [1], Amir A. Saei [1,2], Mike Aoun[3], Noah Moruzzi[4], Ana Coelho[3], Niels Leijten [5], Magnus Nordenskjöld[6,7,8], Patrick Micke [9], Diana Maltseva [10], Alexander G. Tonevitsky[10,11], Vincent Millischer[7,12,13], J. Carlos Villaescusa[14,15], Sandeep Kadekar[16], Massimiliano Gaetani [1,17,18], Kamilya Altynbekova[19], Alexander Kel[19], Per-Olof Berggren[4], Oscar Simonson[16,20], Karl-Henrik Grinnemo[16,20], Rikard Holmdahl [3], Sergey Rodin [1,16,20 ✉] & Roman A. Zubarev [1,21,22 ✉]

Detailed characterization of cell type transitions is essential for cell biology in general and particularly for the development of stem cell-based therapies in regenerative medicine. To systematically study such transitions, we introduce a method that simultaneously measures protein expression and thermal stability changes in cells and provide the web-based visualization tool ProteoTracker. We apply our method to study differences between human pluripotent stem cells and several cell types including their parental cell line and differentiated progeny. We detect alterations of protein properties in numerous cellular pathways and components including ribosome biogenesis and demonstrate that modulation of ribosome maturation through SBDS protein can be helpful for manipulating cell stemness in vitro. Using our integrative proteomics approach and the web-based tool, we uncover a molecular basis for the uncoupling of robust transcription from parsimonious translation in stem cells and propose a method for maintaining pluripotency in vitro.

[1] Chemistry I, Department of Medical Biochemistry and Biophysics, Karolinska Institute, Stockholm 17177, Sweden. [2] Department of Cell Biology, Harvard Medical School, Boston, MA, USA. [3] Division of Medical Inflammation Research, Department of Medical Biochemistry and Biophysics, Karolinska Institute, Stockholm 17177, Sweden. [4] The Rolf Luft Research Center for Diabetes and Endocrinology, Department of Molecular Medicine and Surgery, Karolinska Institute, Stockholm 17176, Sweden. [5] Biomolecular Mass Spectrometry and Proteomics, Bijvoet Center for Biomolecular Research and Utrecht Institute for Pharmaceutical Sciences, Utrecht University, Padualaan 8, Utrecht 3584 CH, The Netherlands. [6] Center for Molecular Medicine, Karolinska University Hospital, Stockholm 171 76, Sweden. [7] Department of Molecular Medicine and Surgery, Karolinska Institute, Stockholm 17177, Sweden. [8] Department of Clinical Genetics, Karolinska University Hospital, Stockholm 171 76, Sweden. [9] Immunology, Genetics and Pathology, Rudbecklaboratoriet, Uppsala University, Uppsala 751 85, Sweden. [10] Faculty of biology and biotechnology, National Research University Higher School of Economics, Myasnitskaya Street, 13/4, Moscow 117997, Russia. [11] Scientific Research Center Bioclinicum, Ugreshskaya str. 2/85, Moscow 115088, Russia. [12] Translational Psychiatry, Center for Molecular Medicine, Karolinska University Hospital, Stockholm 171 76, Sweden. [13] Department of Psychiatry and Psychotherapy, Medical University of Vienna, Vienna 1090, Austria. [14] Neurogenetic Unit, Department of Molecular Medicine and Surgery, Karolinska University Hospital, Stockholm 171 76, Sweden. [15] Stem Cell R&D—TRU, Novo Nordisk A/S, Måløv, Denmark. [16] Department of Surgical Sciences, Uppsala University, Uppsala 752 37, Sweden. [17] Chemical Proteomics Core Facility, Division of Physiological Chemistry I, Department of Medical Biochemistry and Biophysics, Karolinska Institutet, Stockholm 171 77, Sweden. [18] Chemical Proteomics, Science for Life Laboratory (SciLifeLab), Stockholm 17 177, Sweden. [19] geneXplain GmbH, Am Exer 19B, 38302 Wolfenbuettel, Germany. [20] Department of Cardio-thoracic Surgery and Anesthesiology, Uppsala University Hospital, Uppsala 751 85, Sweden. [21] Department of Pharmacological & Technological Chemistry, I.M. Sechenov First Moscow State Medical University, Moscow 119146, Russia. [22] The National Medical Research Center for Endocrinology, Moscow 115478, Russia. ✉email: sergey.rodin@surgsci.uu.se; roman.zubarev@ki.se

Understanding transitions from one cell type or state to another is in the focus of modern molecular biology. Cell-type transitions play a central role in disease modelling, tissue engineering, and regenerative medicine[1–4]. These transitions are enacted by profound changes in protein composition of the cells. In general, proteins regulate their activity by changes in either expression or structure, the latter is often triggered by post-translational modifications (PTMs)[5]. Structural changes frequently lead to alterations in protein solubility, which is emerging as one of the most important parameters modulating protein function[6]. To get a new insight into pluripotency, we employed a plurifaceted experimental design combining expression proteomics with the proteome-wide integral solubility alteration (PISA) assay[7] to compare pluripotent cells with their isogenic progenies and parental cells as well as with allogeneic cells. PISA is a high throughput version of thermal proteome profiling (TPP) or CETSA-MS[8,9]. Thermal profiling performs proteome-wide assessment of protein solubility interpreted as their thermal stability, via the melting curves acquired across a temperature range using multiplexed proteomics[10]. This method adds another dimension to the standard proteomics approaches, which are largely based on quantification of protein abundances and/or the occupancies of their PTMs. TPP is able to detect small changes in solubility or thermal stability of proteins occurring due to the binding of small molecules to proteins in living cells and cellular lysates, post-translational modifications[11], and large macromolecular complex dynamics[12], for example during transitions through the eukaryotic cell cycle[13,14]. Recently, TPP has been used to measure and compare the protein melting temperatures across thirteen different species; the study demonstrated that genomic alterations can affect protein thermal stability[15]. Importantly, changes in protein solubility/stability are found to be complementary to changes in protein abundances[7]. The combined proteomic approach called PISA-Express implemented here uses just two samples per replicate analysis of a cell type (Fig. 1a, b) to assess both the protein abundance and solubility. In contrast, TPP alone usually requires analysis of 10 samples[16]. Following the established tradition, we will interpret the measured solubility alterations as thermal stability changes, bearing in mind that the relation between solubility and stability is actually more complex[17].

In this study, we present a method for simultaneously measuring protein expression and thermal stability after cell-type transition as well as a multidimensional visualization tool, ProteoTracker, based on Sankey diagrams, to study proteome changes without dimension reduction. (Supplementary Fig. 1a, b) (available at http://www.proteotracker.genexplain.com). By analyzing the Sankey diagrams, we detect alterations in protein properties after transition between PSCs and other cell types and map the altered proteins to various pathways and compartments, such as chromatin remodeling, DNA replication, cytoskeleton, cell adhesion, glucose metabolism, and ribosomes. This combined analysis provides a more detailed view of protein behavior that is difficult to obtain from transcriptomic, expression proteomic or TPP analysis only. Lastly, we show that SBDS protein, which is involved in ribosome biogenesis, plays a role in stem cell maintenance and differentiation, and could be targeted to modulate pluripotency in vitro.

## Results

**Protein thermal stability and expression changes during cell-type transitions.** In order to acquire cell lineages that are relevant for regenerative medicine, we reprogrammed hFFs into iPSCs (Supplementary Figs. 2 and 3) and differentiated the latter into EBs using an undirected method (Methods). EBs have been used as a model of early developmental specification and as a starting point in numerous protocols for differentiation of specific cell types, thus representing a valuable tool to study general differentiation processes in pluripotent stem cells[18–27]. We also included to the study a well-known human ESC line H9 as a standard of pluripotency that is allogeneic to the hFFs and iPSCs. Colon cancer cell line RKO was added to the analysis to provide a qualitative and quantitative point of comparison for changes in protein properties occurring between cells from the same parental lineage and sex (iPSCs, hFF and EBs), and cells harboring chromosome rearrangements, various point mutations, and a different sex (RKO)[28]. In addition, RKO cells divide fast[29] and at a similar rate or faster than iPSCs, allowing one to correct at least partially for potential cell division rate bias (Supplementary Fig. 4a). For simplicity, we used the terms "differentiated cells" to designate all the nonpluripotent cell types used in this study. The use of five cell lineages allowed us to fit the full replicate of the PISA-Express analysis within a single proteomics experiment by multiplexing ten tandem mass tags (TMT10). One tenths of every sample, including the replicates, was pooled and labeled by TMT11, and then added to each TMT10 multiplexed sample as a linker, i.e., internal standard (Fig. 1c). Subsequently, the obtained samples were fractionated and analyzed by LC-MS/MS. Thus, in one multiplexed analysis we obtained information on three dimensions, or facets, namely, thermal stability, expression, and degree of differentiation (transiting from pluripotent cells via partially differentiated EBs to fully differentiated hFFs).

In three replicate analyses, we obtained stability and expression data for 9509 proteins, including scarcely expressed master pluripotency markers OCT4 and NANOG. In total, 7778 proteins passed our selection criteria of being identified by at least two unique peptides with no missing values in any replicate and cell type (Supplementary Data 1). For each protein, we calculated the PISA parameter Sm for protein stability where larger Sm value indicates higher thermal stability and vice versa. We also calculated the Exp value for protein expression as the average between three biological replicates (Fig. 1d). For consistency, in both cases the input parameter was the ratio of the TMT reporter ion abundance to that of the linker.

To validate our findings, we increased the proteome depth and sequence coverage in protein expression analysis during differentiation of iPSC and ESC into EBs by using strong detergent (sodium dodecyl sulfate) and quantified 11,451 proteins with 9565 proteins passing our selection criteria (Supplementary Data 2). Additionally, we acquired protein expression datasets on HT29 cells and neurons differentiated from iPSCs hi12, as well as hFF, hi12, and EBs for comparison. The datasets encompass 9000 proteins, of which 5478 passed our selection criteria (Supplementary Data 2), as described in Methods.

**Protein thermal stability distinguishes cell types independently from protein expression.** To investigate how well the Sm and Exp parameters distinguish cell types, we performed principal component analysis (PCA) for all five cell lineages. The PCA plots showed little separation between the iPSCs and ESCs, both for protein stability and expression (Fig. 2a, b). In general, the 1st component in both PCA dimensions aligned with the degree of differentiation of the cells, positioning hFF on the one side and the two PSCs on the other side on each plot. The EBs, which arose from random differentiation and contained some proportion of pluripotent and progenitor cells[30], were found in an intermediate position. RKO cells were most distant from PSCs on the stability plot but in an intermediate position on the expression plot.

Proteins of iPSCs did not correlate with those of hFF, RKO, or EBs in either stability or expression, while the Pearson correlation coefficients with ESC proteins were 0.85 and 0.86, respectively

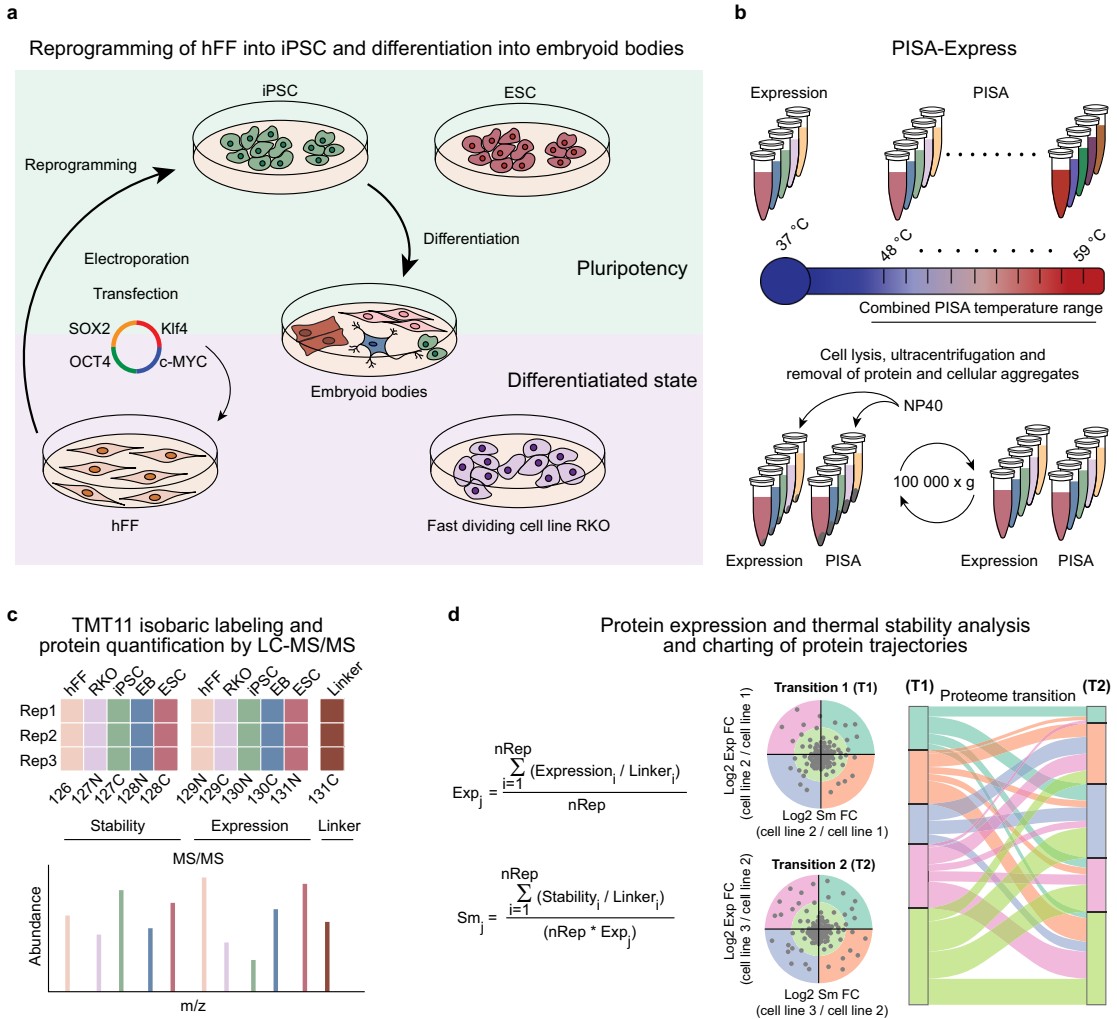

**Fig. 1 Protein thermal stability and expression changes in cell-type transitions. a** Cells of different types in were grown simultaneously ($n = 3$ biologically independent samples per cell type). **b** Cells were collected and heated in a narrow temperature range, with one sample incubated at 37 °C for expression measurements, protein aggregates were eliminated by ultracentrifugation and the soluble proteins were digested. **c** 1/10 portions of each sample were integrated into a pooled sample and digests were labelled by TMT11 and multiplexed. **d** The same pooled sample was used in all multiplexed sets for normalization or reporter ion abundances PISA parameter (Sm) and expression fold change (Exp) were calculated, and protein trajectories during cell transitions were charted using a Sankey diagram.

(Supplementary Fig. 4c, d). For each protein, fold changes (FCs) of Sm and Exp were calculated compared to those of the iPSC population, as well as the standard deviations of these values across the whole dataset. The mean standard deviations of Sm and Exp were also calculated for all proteins. The number of statistical outliers exhibiting more than three mean standard deviations was similar for both expression and stability (Fig. 2c).

Since point mutations can alter protein thermal stability[15,31–33] and expression[34–36], we expected to see the largest differences in RKO compared to other cell types. However, to our surprise, hFF (male) had similar numbers of outliers in stability and higher number of outliers in protein expression than RKO (female) when both were compared to iPSCs (male). On the other hand, ESCs (female) had the lowest number of outliers and showed almost no changes compared to iPSCs (male). These observations suggest that such characteristics as cell identity and/or culture conditions contribute more to the differences in protein properties than the genetic makeup of the cells.

Lastly, we did not observe a significant correlation between the FCs in stability versus expression for individual proteins[7,13,14], confirming that changes in these two analytical dimensions are

independent when comparing different types of cells (Supplementary Fig. 4b). Although the maximum FC amplitude was higher in expression than in stability (Fig. 2c), in PCA the separation between different cell types were similar in the two analysis facets.

**ProteoTracker maps protein trajectories in cell-type transitions.** Protein trajectory between two cell types is determined as a vector in the two-dimensional space of protein stability and expression, with one cell type providing the origin (Fig. 2d). Each protein trajectory falls into one of the five sectors denoted from A to E (Fig. 2e). Sector A included trajectories of proteins that are thermally stabilized and upregulated in the final cell type in comparison with the origin. Similarly, sectors B, C, and D encompass trajectories of stabilized and downregulated, destabilized and downregulated and destabilized and upregulated proteins, respectively. All protein trajectories in sectors A–D have a combined (via Fisher formula[37]) $p$-value $< 0.05$ while all other trajectories were considered statistically insignificant and assigned to sector E that includes the origin (Fig. 2e). In a unidirectional cell transition between three cell types, ProteoTracker maps the

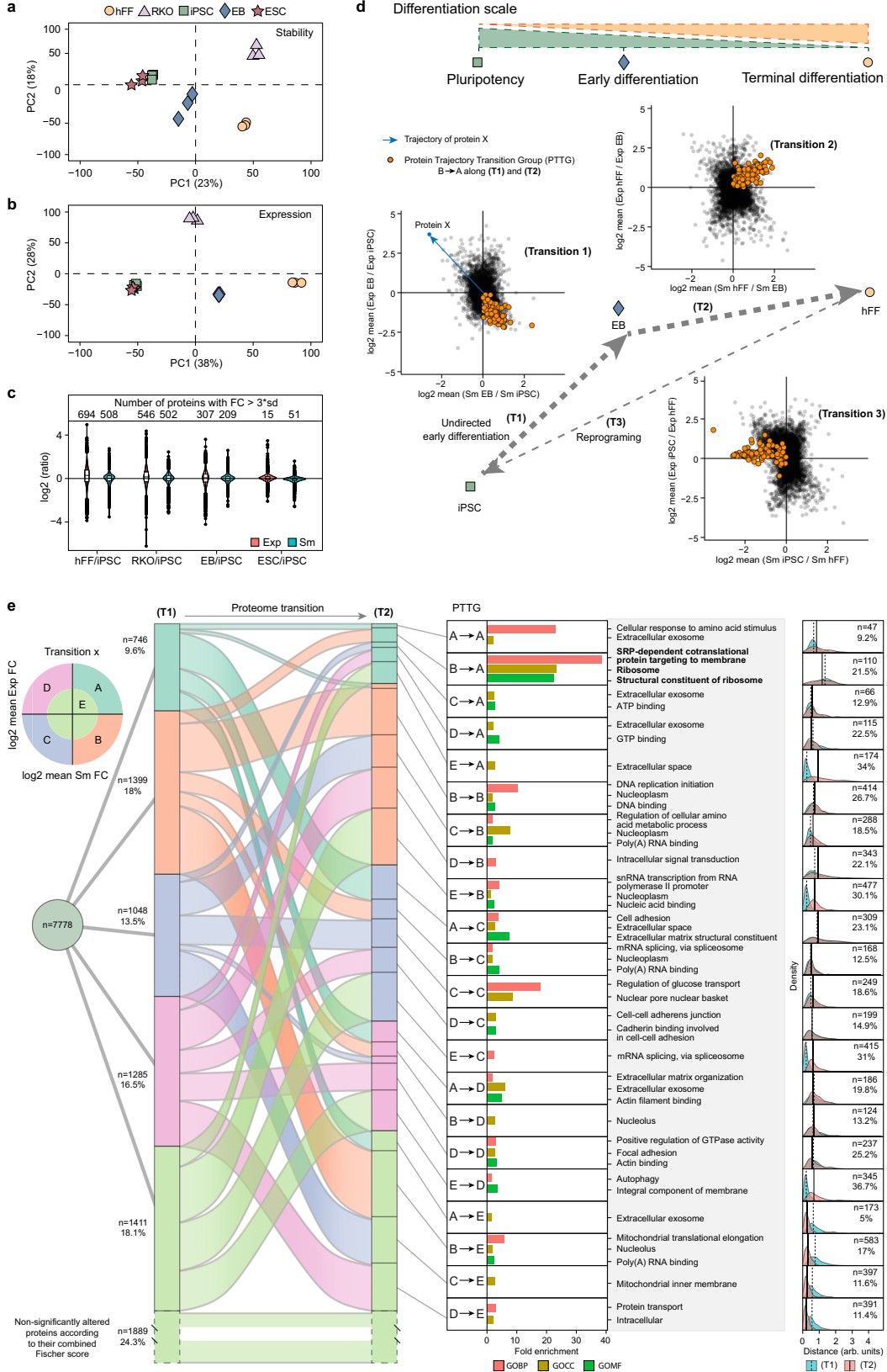

evolution of protein trajectories using a Sankey diagram, in which each of the sectors A–E assigned to a protein trajectory in the first transition transforms to the sector A–E assigned to the trajectory of the same protein in the second transition (Fig. 2e). As an example, in the transition iPSC→EB→hFF, a protein that is stabilized and downregulated (sector B) in iPSC→EB as well as destabilized and upregulated (sector D) in EB→hFF will be classified as a B to D type.

Using this methodology, we studied the proteome conversion from iPSC to hFF via EB as an evolution of two trajectories (Tn),

**Fig. 2 Charting protein trajectories during cell-type transitions using combined protein stability and expression analysis. a** PCA plot of protein stability (Sm) in hi12 iPSC, H9 ESC, EB, hFF, and RKO cells. **b** PCA plot of protein expression (Exp) in each cell type. **c** Violin plots showing the distribution of Sm and Exp FC of each cell line against iPSC and number of proteins with a log2 fold change (FC) in stability and expression exceeding three standard deviations in each cell line compared to iPSC. Horizonal line in the boxplots represent the median, 25th and 75th percentiles and whiskers represent measurements to the 5th and 95th percentiles. **d** Protein trajectories during cell-type transitions were defined as positions in a 2D plot of log2 of FCs for Sm and Exp compared to the original type. Transition (T1) was from pluripotency (iPSC) to early differentiation (EB), and (T2) from EB to terminal differentiation (hFF). **e** Sankey diagram based on assigning to a significant sector (A–D) a protein trajectory in (T1) and (T2) when the combined *p*-value for changes in stability and expression was <0.05, and to an insignificant sector E otherwise. Each protein trajectory transition group (PTTG) of proteins undergoing transition from a sector X in (T1) to a sector Y in (T2) (25 PTTGs in total) was submitted to a Gene Ontology (GO) enrichment analysis with all quantified proteins as background. For each sector in PTTG, the density distribution of the distances on the 2D plot in (T1) and (T2) is plotted and the percentage of proteins transiting from any sector in T1 to sector Y is calculated, and their number is given. Source data are provided as a Source Data file.

EB versus iPSC (T1), and hFF versus EB (T2), with the general direction from pluripotency to terminal differentiation, bearing in mind that EBs are not a fixed cell type but are comprised of a heterogeneous population of pluripotent and progenitor stem cells as well as cells with various degrees of differentiation within many lineages. Gene Ontology (GO) enrichment analysis was performed on each protein trajectory transition group (PTTG) that encompassed proteins undergoing transition from one defined sector to another, for instance from A to A, B to A, etc. (Fig. 2d, e). For each protein and transition, the trajectory length was calculated as a distance to the origin, and their distributions for the corresponding PTTG were plotted to compare the magnitudes of changes in protein properties. The mean distances are shown as vertical lines in the distribution plots (Fig. 2e, right), while the number of proteins and the percentage of proteins in the final sector of each PTTG are also given.

The diagram shows that more than 75% of the proteome significantly changed either expression or stability during the two transitions. The most populated significant (i.e., excluding the sector E) PTTG was B to B, which according to GO enrichment analysis encompasses DNA replication initiation, nucleoplasm and DNA binding. PTTG B to A shows the highest mean distance of trajectories in both T1 and T2 and the highest fold enrichment in GO analysis, with all three most enriched GO terms related to ribosome.

**ProteoTracker reveals progression of molecular pathways during PSC differentiation.** Next, we investigated the progression of key pathways that were enriched in the GO analysis of protein transitions during differentiation of PSCs into EBs. The mean protein distance on T1 in each pathway was normalized such that the distance from the pluripotency state (iPSC) to the terminal differentiation state (hFF) was equal to one unit. Thus, normalized mean magnitude of each pathway for the progression from iPSC to EB was then depicted (Fig. 3a). For cellular amino acid metabolic process, this magnitude of progression was larger than unity, suggesting a forth-and-back modulation of this pathway during differentiation. The pathways related to glycolysis, oxidative phosphorylation, mitochondrial inner membrane, and ribosomal components showed progression close to unity, suggesting the important role of these pathways in early differentiation. The DNA repair, chromatin remodeling, and DNA replication pathways showed much lower progression, hinting that metabolic reprograming precedes the DNA- and chromatin-related changes during differentiation.

To interrogate the data at a different angle, we calculated the position of EBs on a two-dimensional plot composed of the 1st PCA components of stability and expression for each pathway, normalized such that the coordinates of iPSCs are (0, 0) and those of hFF are (1, 1) (Fig. 3b). Strikingly, on this plot the structural component of ribosome was off-scale for expression, albeit close to 0.5 in stability. This hinted that ribosomal proteins' expression

might play a triggering role in PSC differentiation (see below). Mitochondrial inner membrane, glycolysis, oxidative phosphorylation, and cellular amino acid metabolic processes showed similar progression magnitudes for both expression and stability. These magnitudes were higher than for DNA repair, DNA replication, and chromatin remodeling, which shared comparable progression values, similar to their proximity on Fig. 3a.

While metabolic switch and activation of mitochondria are required for cell differentiation[38,39], so is the closure of the open chromatin state in PSCs[40]. However, the timing of these two molecular events has not been fully elucidated. Here, hFF and RKO cells exhibited higher rate of oxidative metabolism and higher respiratory capacity than PSCs (Supplementary Fig. 5c–e). Additionally, EBs had similar expression and stability profiles of proteins involved in glycolysis and oxidative phosphorylation to those of hFF and RKO cells, suggesting that the cells EBs are composed of, at this differentiation stage, mostly rely on oxidative phosphorylation while PSCs uses glycolysis (Supplementary Fig. 5a, b). PSCs are known to have a lower proportion of heterochromatin than differentiated cells rendering the chromatin of pluripotent cells chromatin to be more loose[40–43]. The open state of chromatin in PSCs is maintained by a specific expression pattern of chromatin-remodeling complexes[40]. Here we detected gradual increase in histone stability and changes in expression and stability of chromatin-remodeling complexes during differentiation of PSCs, showing that chromatin in EBs is on average not yet at a fully compacted state characteristic of terminally differentiated cells (Supplementary Fig. 5f–k). Thus, our analysis suggests that metabolic reprograming from glycolysis to oxidative phosphorylation might happen earlier than chromatin rearrangements during early differentiation. This is in line with previous predictions[44].

**Ribosomal proteins increase in stability during differentiation of PSCs.** Since 'the ribosome' was one of the most enriched GO terms in the transition analysis (Fig. 2e) and an outlier in the progression analysis (Fig. 3b), we investigated ribosomal proteins in detail. These proteins were significantly stabilized in differentiated cells compared to PSCs (Fig. 3c–g). Expression of ribosomal proteins was somewhat similar in PSCs, hFFs and RKO, but significantly lower in EBs (Fig. 3c–g). We also performed TPP of hFFs in comparison with two iPSC lines, hi11 and hi13, and confirmed the increased thermal stability of ribosomal proteins in differentiated cells (Supplementary Fig. 6a, b and Supplementary Data 3). Notably, there was no apparent correlation between the ribosomal protein expression and their thermal stability.

**Ribosomal proteins stabilization during differentiation stems from changes in ribosomal structure.** We hypothesized that the ribosomal proteins' stabilization during differentiation of PSCs originates from a difference in the structure of ribosomes rather

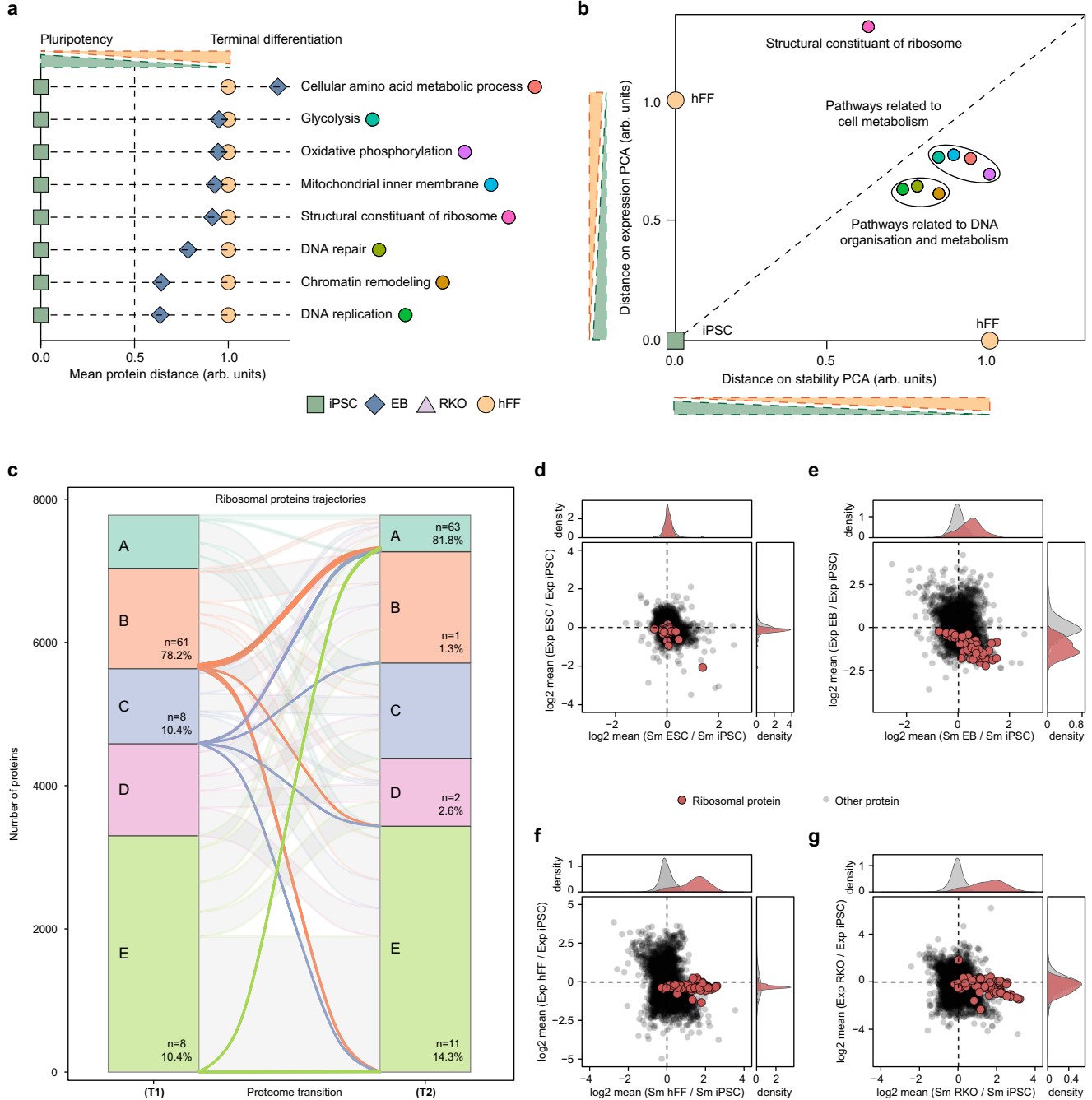

**Fig. 3 ProteoTracker determines timing for molecular events and shows that ribosome is the most affected protein complex during differentiation of PSCs. a** Differentiation scale determined by the mean distances on T1 and T3 normalized by T3 for selected pathways in iPSC, EB, and hFF. **b** Differentiation scale determined by distances to iPSC on PCA of the mean Sm and Exp of proteins in each selected pathway in iPSC, hFF, and EB normalized by the distance between hFF and iPSC. **c** ProteoTracker analysis of ribosomal proteins trajectories in transitions (T1) and (T2). **d-g** Two-dimensional plots of mean Sm and Exp FCs in H9 ESC, EB, hFF, and RKO against hi12 iPSC. Source data are provided as a Source Data file.

than from the changes in intracellular environment. To test this hypothesis, we performed TPP in cell lysates, in which intracellular environment affects protein melting behavior to much lesser degree than in intact cells[12]. Ribosomal proteins had higher average melting temperatures (Tm) in the hFF lysates than in iPSC lysates, confirming the results in living cells (Supplementary Fig. 6c and Supplementary Data 3) and supporting our hypothesis.

**iPSCs have lower proportion of functional ribosomes than RKO.** To investigate the difference in ribosomal structure in

different cell types, we fractionated cell lysates from iPSCs and RKO cells using a 10–50% linear sucrose gradient while recording the absorption at 245 nm. iPSCs demonstrated more abundant 60S subunits (immature ribosomes) with fewer 80S subunits (mature ribosomes) and polysomes compared to RKO cells (Fig. 4a, b). The fractions corresponding to the free proteins, 40S and 60S ribosomal subunits, mature 80S ribosomes, light polysomes and heavy polysomes were analyzed by proteomics, which did not reveal significant differences at either protein or peptide levels (Supplementary Fig. 6f–j and Supplementary Data 2). Thus, the higher thermal stability in differentiated cells did not originate

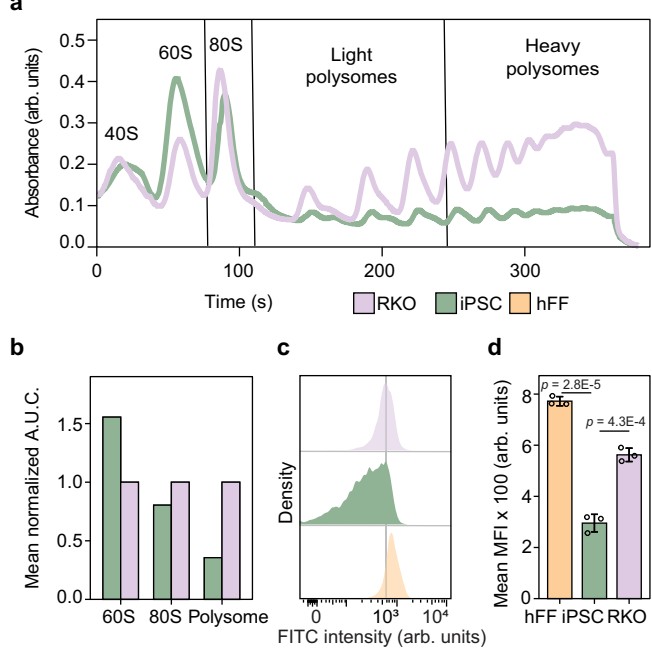

**Fig. 4 iPSCs have lower polysome content and protein synthesis rate than differentiated cells. a** Ribosome density profile on a 10–50% sucrose gradient of hi12 iPSC lysates against that of RKO cells. **b** Analysis of the area under the curve (A.U.C.) of several fractions of the ribosome profile ($n = 2$ biologically independent samples). **c** Measurement of OPPuro incorporation in hi12 iPSC, hFF, and RKO. The gating strategy is described in Supplementary Fig. 6k. **d** Analysis of mean fluorescence intensity (MFI) from OPPuro incorporation in hi12 cells, hFFs and RKO cells ($n = 3$ biologically independent samples for each cell line). P-values were calculated using a two-sided Student t-test. $p < 0.05$ were considered as significant. Error bars represent ±the standard deviation of the mean. Source data are provided as a Source Data file.

from a difference in ribosomal proteins' stoichiometry. There are also doubts that it was caused by a difference in PTMs, as the latter would reduce the level of unmodified peptides, which may result in altered expression read-outs. Translating ribosomes and particularly polysomes appear to be more thermally stable than immature 40S, 60S subunits, or free ribosomal proteins[12]. Thus, higher average thermal stability of the ribosomal pool in somatic cells is likely due to a higher proportion of fully assembled ribosomal complexes and polysomes.

**iPSCs exhibit lower levels of translation than somatic cells**. RKO and iPSCs cells showed similar expression levels of ribosomal proteins, while RKO cells had a higher proportion of functional ribosomes (80S and polysomes). Taken together these results suggest that iPSCs have a lower protein synthesis rate than differentiated cells, in line with previous reports[45]. Reduced protein synthesis capacity due to a lower proportion of functional ribosomes has been shown for *Saccharomyces cerevisiae*[46]. To assess the global protein synthesis rates in iPSCs, RKO, and hFF cells, we used incorporation of a puromycin analogue into newly translated proteins (Click-iT® Plus OPP Protein Synthesis Assay). As expected, the pluripotent cells exhibited significantly reduced mean protein synthesis rate in comparison with that of hFF and RKO cells (Fig. 4c, d and Supplementary Fig. 6k).

**SBDS is downregulated in iPSCs in comparison with somatic cells**. To understand the molecular basis for the decrease in stability of ribosomal proteins in pluripotent cells and its link to the

reduced protein synthesis rate, we investigated the expression of proteins involved in ribosome biogenesis in several cell lineages, including iPSCs, neuronal cells (Supplementary Fig. 7a, b) and EBs differentiated from the iPSCs, parental hFFs, HT29, and RKO cells. We determined the expression levels for 5478 proteins that passed our criteria across all cell types. SBDS protein showed the lowest mean expression in iPSCs compared to differentiated cell types, being the only ribosome biogenesis-related protein downregulated in iPSCs compared to any tested differentiated cell line, and anticorrelating in expression levels with other ribosome biogenesis factors (Fig. 5a–c Supplementary Data 2). SBDS protein expression profiles in hFF and iPSC were corroborated by microarray analysis of mRNA showing that SBDS mRNA is present in lower quantities in iPSCs than in hFF (Supplementary Data 4). SBDS presence is necessary in the late steps of 60S ribosomal subunit maturation, critical for the assembly of translation-competent ribosomes[47,48]. SBDS cooperates with EFL1 for ejecting EIF6 from immature 60S subunit that leads to association of a 60S subunit and a 40S subunit into a functional 80S ribosome[49]. Therefore, the deficit of SBDS in iPSCs may hamper the assembly of mature ribosomes and formation of polysomes resulting in the observed lower protein synthesis rate.

**SBDS knockdown (KD) reduces protein translation in hFFs**. To test if downregulation of SBDS reduces the protein synthesis rate, we treated hFFs with siRNA against SBDS and scrambled siRNA (control). SBDS knockdown (KD) was confirmed with LC-MS/MS (Supplementary Data 2), revealing a downregulation of SBDS to 50 and 45% after 2 and 4 days of treatment, respectively (Supplementary Fig. 6d, e). Although the remaining SBDS levels were still higher than in iPSCs, SBDS KD hFFs showed significant reduction in the global protein synthesis rate in comparison with hFFs treated with the control siRNA (Fig. 5d).

**SBDS exhibits opposite expression pattern compared to pluripotency markers and other ribosome biogenesis factors during differentiation of PSCs**. To study SBDS expression during PSC differentiation, we performed LC-MS/MS analysis of iPSCs and EBs formed from the iPSCs at different time points of the differentiation process (Fig. 6a and Supplementary Data 2). SBDS expression steadily increased starting from day 6 of the iPSCs differentiation. Importantly, SBDS expression had significant opposite expression trend compared to that of the master pluripotency markers OCT4 and NANOG (Fig. 6b), as well as to the mean expression of other ribosome biogenesis factors and ribosomal proteins over the 9 days long course of the EB induction (Fig. 6c). Real-time quantitative PCR (qRT-PCR) analysis confirmed the increase in SBDS transcription and the decrease in *OCT4* and *NANOG* transcription at day 9 of the iPSCs differentiation (Fig. 6d).

Recently, Kanton et al. have analyzed PSC differentiation into cerebral organoids using single-cell transcriptomics and provided a tool for reconstruction of differentiation trajectories from pluripotency through neuroectoderm and neuroepithelial stages followed by divergence into various neuronal fates[50]. Using the tool for the analysis of human ESC line H9 and human iPSCs line 409b2 differentiation, we visualized expression of OCT4, NANOG, and SBDS along the differentiation trajectories and confirmed the downregulation of SBDS expression in OCT4- and NANOG-positive cells in comparison with that in differentiating and differentiated neuronal cells (Supplementary Fig. 7c–e).

To further validate SBDS deficit in PSCs, we formed EBs from human ESC lines H9 and HS980, and compared protein expression profiles in the human ESCs, EBs at the 9th day of the differentiation process, as well as in hFFs (Supplementary

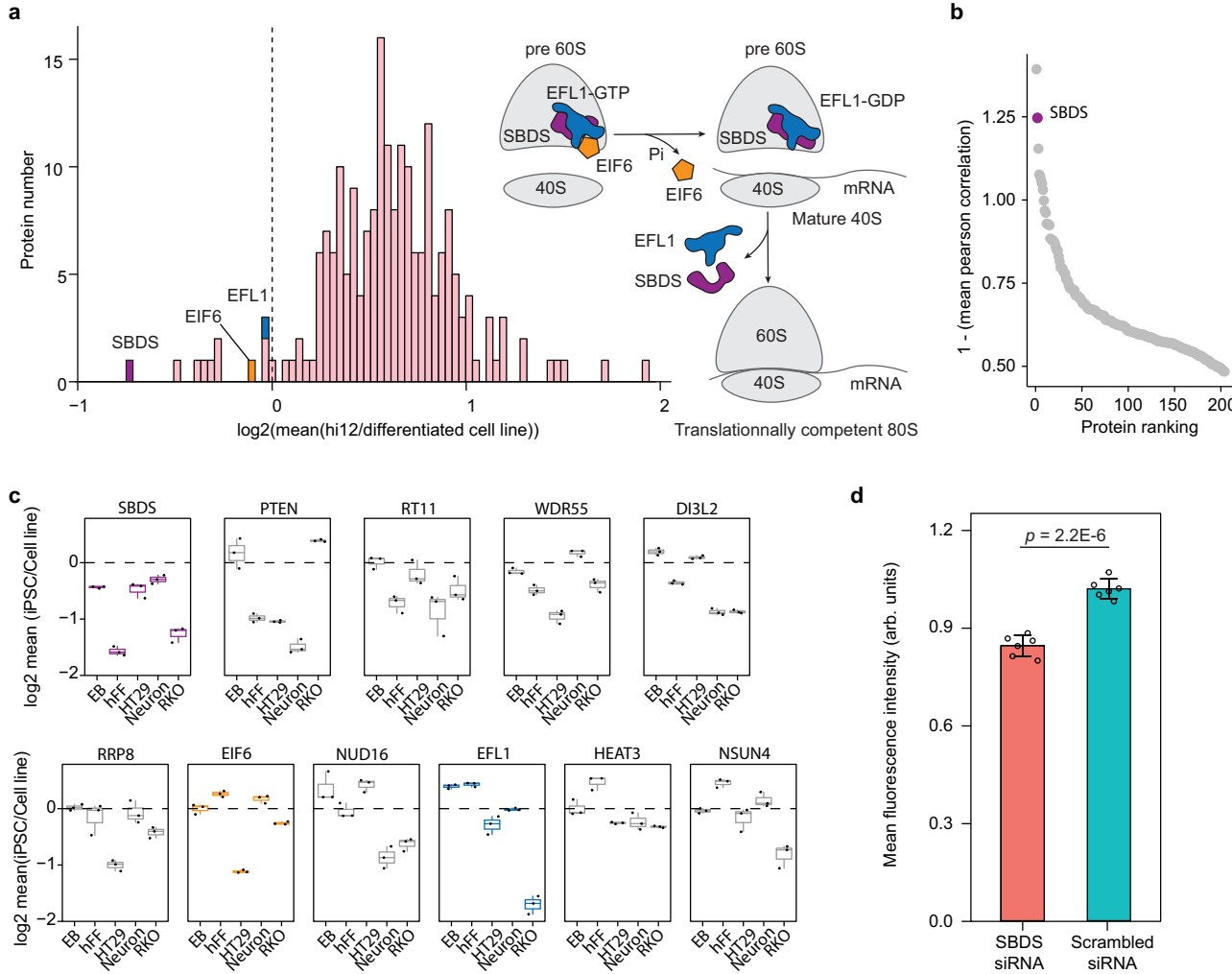

**Fig. 5 SBDS controls protein synthesis rate in iPSCs. a** Expression of proteins involved in ribosome biogenesis in hi12 iPSCs compared to that in parental hFFs, EBs, and neuronal cells differentiated from hi12 iPSCs, HT29, and RKO cells ($n = 3$ biologically independent samples). **b** Ranking of all ribosome biogenesis factors according to the reversed mean correlation of their expression against each other individual ribosome biogenesis factors' expression in all the cell lines mentioned above. **c** Expression of the ten most downregulated ribosome biogenesis factors in each cell line compared to that in hi12 iPSC. Boxplots of SBDS, EIF6, and EFL1 proteins are colored in purple, orange, and blue, respectively, while other proteins are colored in gray ($n = 3$ biologically independent samples). Horizonal line in the boxplots represent the median, 25th and 75th percentiles and whiskers represent measurements to the 5th and 95th percentiles. **d** OPPuro incorporation measurement in SBDS siRNA and scrambled siRNA (control) treated hFFs 3 days after the treatment, data were normalized to the mean fluorescence intensity in the scrambled siRNA control ($n = 6$ biologically independent samples). Error bars represent ±the standard deviation of the mean. *P*-values were calculated using a two-sided Student *t*-test. $p < 0.05$ were considered as significant. Source data are provided as a Source Data file.

Data 2). SBDS levels were significantly higher in EBs and hFFs compared to those in ESCs; the differences were on par with those observed in the experiments with differentiation of iPSCs (Supplementary Fig. 8a, c, d, e and Supplementary Data 2). Moreover, SBDS expression in human ESCs and their EBs also showed opposite expression patterns compared to those of the ribosomal proteins and ribosome biogenesis factors as well as OCT4 and NANOG.

**SBDS KD maintains pluripotency and SBDS knockin decreases the expression of pluripotency markers in PSCs.** Finally, EBs were also treated with siRNA against SBDS and control scrambled siRNA at day 6 of the EB induction and analyzed at day 9 (Fig. 6a). The qRT-PCR analysis showed significant downregulation of *SBDS* mRNA levels and significant upregulation of *OCT4* and *NANOG* mRNAs in SBDS KD EBs in comparison with controls (Fig. 6e). SBDS siRNA treatment decreased SBDS

expression to 61% as well as globally reduced expression of proteins that, according to GO annotation, are involved in anatomical structure and organ development processes. This result confirmed the decreased ability of iPSCs to differentiate upon SBDS downregulation (Fig. 6f, Supplementary Data 2). To assess the reproducibility of the significance cutoff that we used, a permutation analysis of the replicates was performed. In 57,390 unique permutations, only two proteins showed significantly altered expression (false positives) using our cutoff, while 155 out of 9565 proteins had significantly altered expression in the analysis of treated samples versus controls.

The above observations were confirmed by transferring iPSCs onto Geltrex™, which is known to induce more spontaneous differentiation than laminin-521[51], and treating the cultures with two different siRNAs against SBDS and two scrambled siRNA controls. The *SBDS* mRNA level was reduced by around 50% in treated cells in comparison with controls (Fig. 7a). Also, SBDS

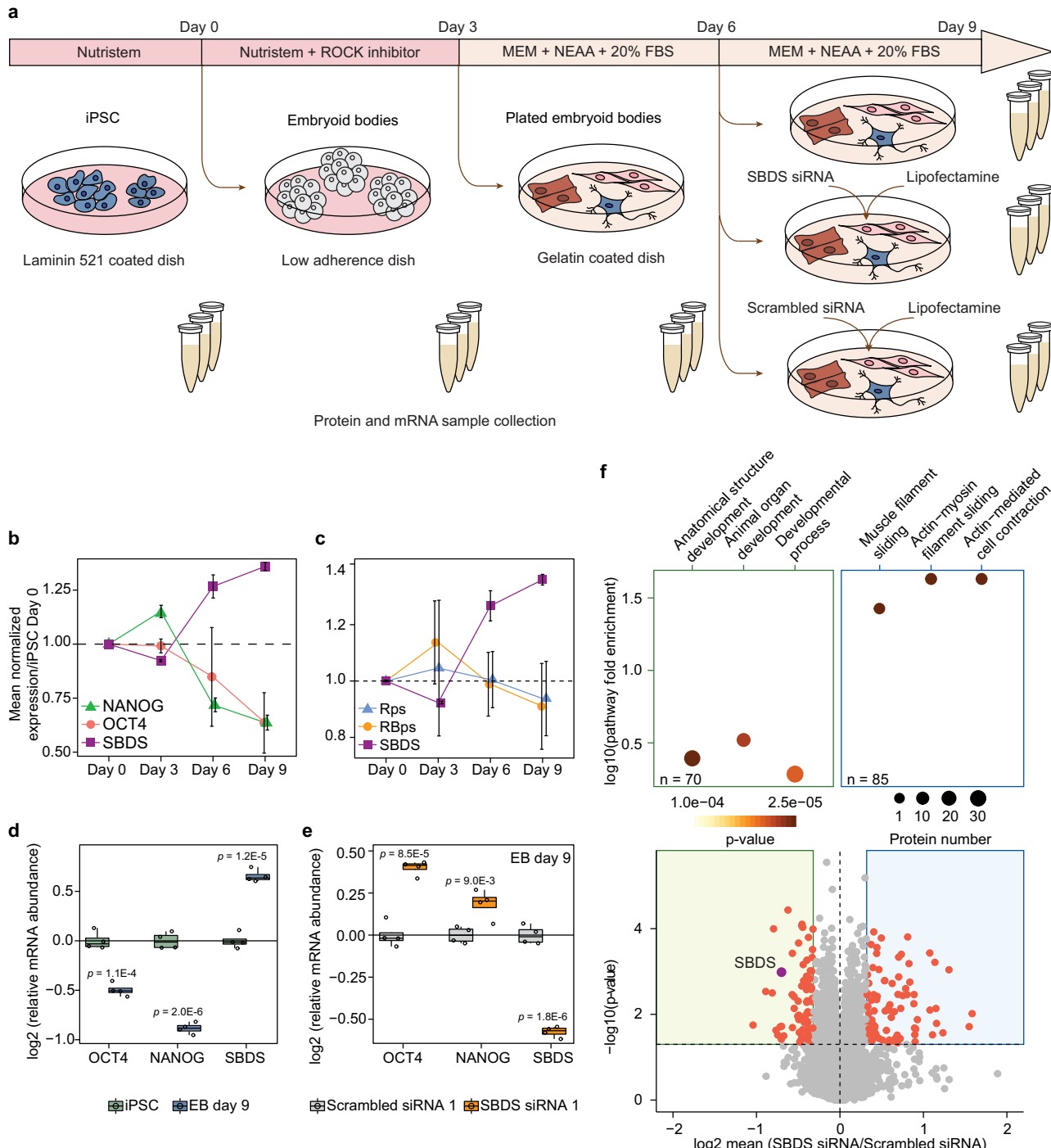

**Fig. 6 SBDS protein is involved in differentiation and pluripotency maintenance in PSCs. a** Schematic representation of PSCs differentiation into EBs and subsequent SBDS siRNA and scrambled siRNA treatments. **b** Protein expression of SBDS, OCT4, and NANOG in EBs at different days of their induction normalized to that in hi12 iPSCs at day 0. **c** Expression of SBDS protein, mean ribosomal proteins expression (Rps) and mean expression of proteins involved in ribosome biogenesis (RBps) encompassing 81 and 237 unique proteins, respectively, in EBs normalized to that in hi12 cells at day 0. Error bars represent ±the standard deviation of the mean. **d** Relative abundance of SBDS, OCT4, and NANOG mRNAs after nine days (D9) of EBs induction versus that in day 0 (D0) hi12 cells measured using qRT-PCR. **e** Relative abundance of SBDS, OCT4, and NANOG mRNAs in day 9 EBs after three days of SBDS siRNA and scrambled siRNA treatments. **f** Relative protein abundance in day 9 EBs after 3 days of SBDS siRNA treatment versus that in the day 9 EBs treated with scrambled siRNA (control); and top three most significantly up or downregulated pathways according to GO annotation ($P < 0.05$ and relative expression <0.8 or >1.25) ($n = 3$ biologically independent samples). $P$-values were calculated using Welsh's $t$-test for proteomics experiment and using a two-sided Student $t$-test for qRT-PCR. $p < 0.05$ were considered as significant. Horizonal line in the boxplots represent the median, 25th and 75th percentiles and whiskers represent measurements to the 5th and 95th percentiles. $n = 3$ biologically independent samples for proteomics experiments and $n = 4$ biologically independent samples for qRT-PCR data. Source data are provided as a Source Data file.

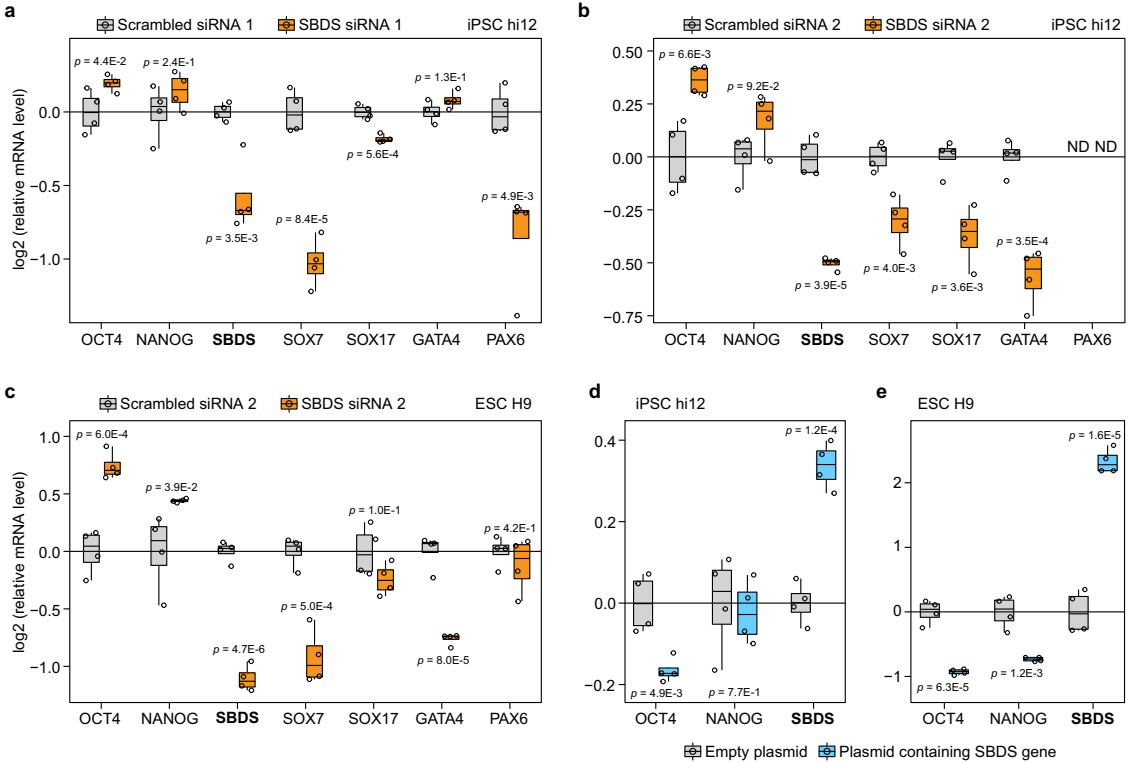

**Fig. 7 SBDS knockdown promotes maintenance of pluripotency and knockin decreases expression of pluripotency markers in PSCs.** Relative mRNAs abundance of pluripotency (OCT4 and NANOG) and lineage specific markers (SOX7, SOX17, GATA4, and PAX6) after 3 days of scrambled siRNA 1 and SBDS siRNA 1 treatments (**a**) or scrambled siRNA and SBDS siRNA 2 treatments (**b**) in iPSCs hi12 and scrambled siRNA 2 and SBDS siRNA 2 treatments in ESCs H9 (**c**) grown on Geltrex™. Relative mRNAs abundance of pluripotency markers (OCT4 and NANOG) after knockin of SBDS protein in hi12 (**d**) and H9 (**e**). Horizonal line in the boxplots represent the median, 25th and 75th percentiles and whiskers represent measurements to the 5th and 95th percentiles. ND nondetected. *P*-values were calculated using a two-sided Student *t*-test. $p < 0.05$ were considered as significant. $n = 4$ biologically independent samples for all experiments. Source data are provided as a Source Data file.

knockdown significantly increased the mRNA level of the master pluripotency marker *OCT4* and, overall, significantly decreased mRNA levels of lineage specific markers including *SOX7* (endoderm), *SOX17* (endoderm), *GATA4* (mesoderm), and *PAX6* (ectoderm) in comparison with the control iPSCs (Fig. 7a, b). Furthermore, SBDS KD in ESCs H9 led to a significant increase in mRNA levels of both *OCT4* and *NANOG* and a significant decrease in mRNA levels of lineage markers *SOX7* and *GATA4* (Fig. 7c). Importantly, both SBDS siRNAs treatment did not significantly alter pluripotent stem cells viability and proliferation (Supplementary Fig. 8b). Lastly, knockin of SBDS protein decreased mRNA levels of *OCT4* in iPSC hi12 (Fig. 7d) and of *OCT4* and *NANOG* in ESCs H9 (Fig. 7e).

## Discussion

Here, we provide the proteome signature resource ProteoTracker comprising a plurifaceted dataset and online software for comprehensive system-wide analysis of thermal stability and expression level changes of proteins and apply this tool for comparing several isogenic and allogeneic cell types including well-defined pluripotent cells. We observed that protein thermal stability distinguished cell types and degrees of differentiation on the same scale as protein expression. The visualization feature of Proteo-Tracker helped interrogating our plurifaceted dataset, mapping changes in protein properties after cell-type transitions on an easily interpretable Sankey diagram. Rather uniquely, Sankey diagrams visualize multidimensional system-wide proteome transitions in form of 2D maps without reduction of the underlying dimensionality. Importantly, our dataset and interface can

be both expanded to include more types of somatic and pluripotent stem cells and be used for analysis of other transitions within and between cell lineages, particularly the transitions between somatic stem cells and their differentiated progeny.

Using the ProteoTracker tool, we studied protein trajectories between isogenic cell types and detected alterations of protein properties in multiple cell compartments and pathways, reflecting fundamental differences in cell physiology and morphology between somatic cells and PSCs. Some of the changes could arise from differences in the growth conditions. Our analysis shows that more than 75% of the proteins present in our dataset vary significantly in expression and/or stability between the pluripotent and differentiated cell types. Particularly, expression and/or stability of histones and chromatin remodelers, including SWI/SNF and Mi2/NuRD families, were altered between PSCs and differentiated cells. This can be attributed to the particular chromatin landscape in PSCs and suggests that our approach is suitable to study chromatin compaction. We also found that EBs already exhibit expression and stability patterns of metabolic enzymes similar to those of differentiated cells before achieving the levels of chromatin compaction characteristic of terminally differentiated cells. Finally, our analysis of timing of activation of the molecular pathways suggests that the metabolic switch is occurring in the early stages of PSCs differentiation, and as it happens before the complete rearrangement of the chromatin, it may be the trigger of the latter process[44].

The most intriguing finding relates to the role of ribosomes in stemness maintenance and cell differentiation. We discovered thermal destabilization of ribosomes in PSCs and gradual increase in the ribosome stability along the differentiation path. Our data

confirmed earlier suggestions that increased thermal stability of ribosomal proteins is linked to a higher proportion of mature ribosomal complexes[12]. Using ribosome density profiling, we showed that PSCs possess lower levels of mature ribosomes than differentiated cells. The decreased ribosome stability in pluripotent cells explains the decreased protein synthesis rate despite high division rate and, in general, higher expression levels of ribosomal proteins and ribosome biogenesis factors than in differentiated cells. Taken together, the provided evidence explains the reported inefficacy of translation in pluripotent cells[45,52].

Using our approach, we also identified SBDS protein as the ribosomal factor that helps maintaining pluripotency in PSCs. It has been shown that SBDS is a key component in the late stage of ribosome maturation and its deficit inhibits association of ribosome subunits into mature 80S ribosomes, thus limiting the translation rate[47]. Here, we showed that differentiation of PSCs is associated with an increase in SBDS abundance and is impaired by SBDS knockdown. The former corroborates the reported increase of translation efficiency during differentiation of PSCs[45,52], while the latter suggests a role of SBDS in inhibiting spontaneous differentiation during the self-renewal of PSCs. Conversely, SBDS knockin was characterized by a decrease in expression of the pluripotency markers. Our data suggest that physiologically low level of SBDS in PSCs helps maintaining pluripotency via control of the protein translation rate by suppression of the ribosome maturation and that on the contrary, higher expression levels of SBDS may promote the exit from pluripotency. This finding also provides an explanation to the putative discrepancy between the elevated transcriptional[41] and lowered translational[45] activities that are known features of PSCs. These results are consistent with the earlier findings that mutation in the *SBDS* gene or SBDS knockdown do not affect long-term self-renewal of PSCs[53].

Mutations in the *SBDS* gene that reduce expression of functional SBDS protein are known to induce an immature ribosome phenotype similar to the one demonstrated here in PSCs and to cause Shwachman-Diamond-Syndrome (SDS)[54]. SDS is a genetic ribosomopathy that causes bone marrow disorders as well as various organ malformations, including skeletal and neuronal disorders in the patients[55,56]. The bone marrow phenotype in SDS affects the ability of hematopoietic stem cells to differentiate rather than reduces the stem cell pool itself[45]. This is analogous to the effect of SBDS knockdown in differentiating cells demonstrated here. Our data further suggest that apart from causing the bone marrow phenotype, the altered SBDS expression also impairs the ability of pluripotent cells to differentiate, thus providing a new insight into other developmental defects observed in patients with SDS.

Several types of somatic stem cells have been shown to restrict their translation rate and to increase it upon differentiation[57–59], suggesting that lowering translation rate is important for maintaining cell stemness. In addition, Kruta et al.[60] have recently shown that inhibition of protein synthesis facilitates maintenance of HSCs in vitro. We speculate that inhibition of SBDS might be a useful universal approach for maintaining stem cells in vitro.

In conclusion, our data on the proteins involved in chromatin remodeling, DNA replication machinery, metabolism, and ribosome stability confirm that this plurifaceted proteomics dataset and the interpretation tool can be used to study stem cell related and other cell-specific phenomena. Here, we only explored a few features of the proteome conversion, leaving a multitude of intriguing findings for further functional exploration and discoveries. This plurifaceted proteomic approach may prove to be a useful tool in finding optimal conditions for proliferation of various cells, including self-renewal of stem cells, designing differentiation and trans-differentiation protocols and mechanistic

studies on particular proteins functions. We expect that PISA-Express proteomics coupled to the analysis of protein trajectory transitions may be broadly used in cell biology.

## Methods

**Cell culture.** All cell lines mentioned below were cultured at 37 °C and 5% $CO_2$ in a humidified Forma Steri-cycle i160 $CO_2$ incubator (Thermo Fischer Scientific) and were grown in the described media unless otherwise specified. HFF (ATCC® CRL-2429™, male), passage 3–6, were grown in IMDM medium (Biowest) supplemented with 10% Gibco™ Fetal Bovine Serum, Qualified, US Origin, Standard (Sterile-Filtered) (Gibco). RKO cells (ATCC® CRL-2577™, female), passage 3–5, were grown in DMEM supplemented with 2 mM glutamine (Lonza) and 10% FBS. HT29 (ATCC® HTB-38™, female), passage 3–5 were grown in RPMI (Lonza) supplemented with 10% FBS. ESCs, wild-type H9 (WiCell WA-09, female)[61], passage 35–40 and HS980, passage 9–25 were grown on LN-521-coated dishes in NutriStem medium (Biological Industries).

**Reprograming and culturing of pluripotent cells.** Human foreskin fibroblasts (hFFs, ATCC® CRL-2429™) were reprogramed in hi11, hi12, and hi13 iPSC lines using CoMiP 4in1 plasmid without shRNA p53 using a method described here[62]. One day prior the procedure, 400 K fibroblasts were plated on a 60 mm plate. Next day, the cells were collected using TrypLE (ThermoFisher Scientific) treatment and electroporated using 9 µg of the CoMIP plasmid. The electroporation was performed using 4D-Nucleofector (Lonza Bioscience) in P2 solution using DT-150 program. After electroporation, the cells were plated in four wells of a six-well plate precoated with laminin-521 (BioLamina). From day 3 till day 5, the fibroblast medium was gradually changed to TeSR™-E7™ (STEMCELL Technologies, Canada) and from day 9 till day 14 to NutiStem (Biological Industries). From day 1 till day 9, the media were also supplemented with 0.2 mM sodium butyrate and 50 µg/mL ascorbic acid. Between day 14 and day 21, the iPSC-looking colonies were mechanically collected and plated into wells of a 96-well plate precoated with laminin-521. The cells were confirmed to express pluripotency markers NANOG (R&D Systems, Cat. No. AF1997, Lot No. KKJ0409101. Dilution 1:10), Sox-2 (R@D Systems, Cat. No. MAB2018, Lot No. KGQ020841. Dilution 1:10), SSEA-4 (Invitrogen, Cat. No. MA1-021-D488, Lot No. TH274753. Dilution 1:200) by immunostaining (Supplementary Fig. 2a, b). Lack of the plasmid integration was confirmed using RT-PCR as described here[62] (Supplementary Fig. 2c). Karyotyping revealed no genetic aberrations in hi11, hi12, and hi13 cells (Supplementary Fig. 2d–f). Pluripotency of hi11, hi12, and hi13 cells was demonstrated in in vivo (teratoma formation in mice) and in vitro (immunostaining of embryoid bodies) experiments (Supplementary Fig. 3a–j). The pluripotent cells were cultured on LN-521 in NutriStem medium (Biological Industries) at 37 °C, 5% $CO_2$ as described here[51]. iPSCs were used at passage between 15 and 25 in the various experiments.

**Immunostaining.** Cells were cultured in 96-well plates (Corning) and fixed using 4% paraformaldehyde, permeabilized by 0.1% Triton X in PBS and blocked by 10% bovine fetal serum (GIBCO Invitrogen Corporation) in PBS containing 0.1% Tween-20 (Sigma–Aldrich) for 1 h at room temperature. The specimens were incubated with primary antibody for 1.5 h at room temperature, washed three times with 0.1% Tween-20 in PBS buffer, incubated with secondary antibody plus 4',6-diamidino-2-phenylindole (DAPI, Molecular Probes) for 1.5 h at room temperature, washed three times with 0.1% Tween-20 in PBS buffer and, finally, washed two times with PBS. The specimens were preserved in a fluorescence mounting medium (Dako, Glostrup, Denmark) at 4 °C and analyzed using Operetta fluorescence microscope (Perkin Elmer).

**FACS analysis.** Cells were removed from cell culture dishes using TrypLE Express, pelleted, fixed with 4% paraformaldehyde, washed with PBS and resuspended into single-cell suspension in FACS buffer (2% fetal bovine serum in Hank's buffer). Incubation with SSEA-4 antibodies was performed for 30 min at room temperature in FACS buffer. Then, cells were washed five times with FACS buffer and placed on ice. Cells were analyzed by FACSCalibur Flow Cytometer (Becton Dickinson, San Jose, CA) using CellQuest software (Becton Dickinson).

**Karyotyping.** Karyotyping of hi11, hi12, and hi13 cells after 16, 23, and 9 passages in culture, respectively, was done using standard Q-banding technique. The cells were treated with 0.1 µg/mL of KaryoMAX (GIBCO Invitrogen) for 5 h. After that, the cells were dissociated with TrypLE Express (GIBCO Invitrogen), pelleted by centrifugation, resuspended and incubated in hypotonic solution (0.0375 M KCl) for 10 min. After the incubation, the cells were pelleted and fixed in 3:1 methanol/acetic acid. Metaphase spreads were prepared on glass slides, G-banded by brief exposure to trypsin and stained with 4:1 Gurr's/Leishmann's stain (Sigma–Aldrich). A minimum of 10 metaphase spreads were analyzed, and an additional 20 were counted.

**Teratoma formation.** Teratoma formation experiments were done by implantation of cells beneath the testicular capsule of young (8 week old) NOD SCID mice. Three animals were used for each cell line. One million cells were injected in a 30%

Matrigel solution in a total volume of 100 μL. Teratoma growth was observed by weekly palpation, and the mice were sacrificed 8–12 weeks after the implantation. The teratomas were fixed, and sections were stained with hematoxylin and eosin. The tissue was microscopically assessed by a pathologist. All animal experiments were performed at the infection-free animal facility of the Karolinska University Hospital Huddinge and approved by the Ethics Committee on Animal Experiment, Sweden (ethical approvals number S198-11 and S31-14). We have complied with all relevant ethical regulations for animal testing and research.

**Embryoid body formation**. Pluripotent cells were dissociated from dishes using treatment with TrypLE Express, pelleted by centrifugation, resuspended in NutriStem (Saveen Werner) with 10 μM of Y-27632 (a ROCK inhibitor) and cultured in suspension on low adhesion plates at 37 °C, 5% CO$_2$ to form EBs. The EBs aimed for immunostaining after 1 week in suspension were plated in gelatin-coated tissue cell culture 96-well plates (Corning) in DMEM medium (GIBCO) supplemented with 20% (vol/vol) fetal bovine serum (GIBCO), 2 mM l-glutamine and 1% (wt/vol) nonessential amino acids (GIBCO) and ascorbic acid 50 μg/ml (Sigma–Aldrich). After additional week in culture, the spread EBs were fixed, stained for markers of all three germ layers with antibodies against smooth-muscle actin (SMA) (Merck-Millipore, Cat. No. A5228. Dilution 1:250), MAP-2 (Millipore, Cat. No. MAB3418, Lot No. LV 1796719. Dilution 1:200), and α-fetoprotein (AFP) (R@D Systems, Cat. No. MAB1368, Lot No. HPH0209081. Dilution 1:100) and analyzed as described above for immunostaining.

**Generation of human neurons**. Human neurons were generated from the hi12 line. hi12 cells were dissociated into single cells using TrypLE Select and transferred onto nonadhesive plastic plates in DMEM/F12 containing N2 (1:100 Gibco Invitrogen, New York, USA, Cat. No. 17502001), as well as the SMAD inhibitors 431542 (Sigma–Aldrich/Merck, Cat. No. 301836-41-9) and LDN-193189 (Sigma–Aldrich, St. Louis, Missouri, USA, Cat. No. 1062368-24-4). For the first 24 h, 10 μM of ROCK inhibitor Y-27632 (Tocris, Cat. No. 1254) was added, leading to the formation of floating cell aggregates. Half the medium was replaced daily. At day 6, the floating aggregates were plated onto tissue cultured plates coated with 0.002% poly-L-ornithine (Sigma–Aldrich, St. Louis, Missouri, USA, Cat. No. 27378-49-0) and 20 μg/mL murine laminin (Sigma–Aldrich, St. Louis, Missouri, USA, Cat. No. 114956-81-9), which led to the attachment of the cells. After 2–3 days, neural rosette structures started to emerge, which were considered neural precursor cells. These were picked manually after 4 days and seeded at a concentration of 150,000 cells/cm$^2$, in DMEM/F12, supplemented with 1% N$_2$, 0.1% B27 (Gibco Invitrogen, New York, USA, Cat. Num. 17504044), SHH (Sonic Hedgehog, 200 ng/ml, R&D), and CT (1 μM CT99021, Sigma). After 7 days of differentiation, SHH and CT were removed from media, and recombinant BDNF (20 ng/mL, R&D), GDNF (20 ng/mL, R&D) and Ascorbic Acid (200 μM, Sigma) were added to the culture media until day 12. Neuronal rosettes and neurons were prepared for immunostaining as described above using rabbit anti-Nestin (Atlas Antibodies, Cat. No. HPA007007. Dilution 1:500), mouse monoclonal anti-ZO-1 (Molecular Probes, Cat. No. 339194. Dilution 1:500), and mouse monoclonal anti-βIII Tubulin (TUBB3) (Promega, Cat. No. G7121. Dilution 1:500).

**Cell division rate measurement**. hi12, hFF, and RKO cells were grown until exponential growth phase and processed using Click-iT™ EdU Cell Proliferation Kit for Imaging (Thermo Fischer Scientific) according to manufacturer's instructions. The results were analyzed using Operetta fluorescence microscope.

**PISA of cell types**. HFFs (passage 5), iPSCs (hi12, passage 21), ESCs (H9, passage 40), EBs (from hi12 passage 19), and RKO (passage 3) cells were grown as described above until 80% confluence and for 21 days for EBs. Then cells were washed with PBS and detached using TrypLE Express. The reaction was stopped by adding fresh medium and the cells were pelleted at 340 × g for 2 min. Cell pellets were rinsed two times with PBS and finally the cells were resuspended in 1 mL of PBS supplemented with protease inhibitors (Roche) and 100 μL of the cell suspension were distributed into 10 PCR tubes per replicate (n = 3 biologically independent samples for each cell line). Cells were heated in a temperature range from 48 to 59 °C and samples corresponding to each replicate were combined together. For each cell line, one sample designated for protein expression measurement was incubated at 37 °C (n = 3 biologically independent samples for each cell line) and processed alongside the pooled samples. NP-40 (Sigma–Aldrich) was added to lyse the cells with a final concentration of 0.4% and cells were further lysed using repeated freeze/thaw cycles. Finally, all lysates were transferred to ultracentrifuge tubes, placed into a Ti 42.2 rotor (Beckman-Coulter) and ultracentrifuged at 100,000 × g for 20 min using an Optima XPN-80 Ultracentrifuge (Beckman-Coulter). Seventy microliters of the supernatant was collected and the same volume of lysis buffer (8 M urea, 20 mM EPPS pH 8.5) was added and the protein concentration was measured using Pierce bicinchoninic acid assay (BCA) protein assay kit (Thermo Fischer Scientific) according to the manufacturer's protocol.

**Thermal proteome profiling**. HFFs, iPSCs (hi11, hi12, and hi13 cell lines), and RKO cells were grown as described above until 80% confluence. Then cells were washed with PBS and detached using TrypLE Express. The reaction was stopped by adding fresh medium and the cells were pelleted at 340 × g for 2 min. Cell pellets were rinsed two times with PBS and finally the cells were resuspended in 1 mL of PBS supplemented with protease inhibitors (Roche) and 100 μL of the cell suspension were distributed into 10 PCR tubes per replicate (n = 2 biologically independent samples for each cell line).

For TPP in cell lysate, hi12 cells and hFFs were grown until 80% confluence, then detached using TrypLE Express (the enzymatic activity was stopped by adding fresh medium). Cells were then pelleted, rinsed two times with PBS and resuspended in PBS supplemented with protease inhibitors. After that, the cellular suspensions were freeze-thawed five times in liquid nitrogen for cell lysis with strong vortexing in between each cycle. The insoluble fractions were pelleted for 5 min at 20,000 × g and then 100 μL of the supernatants were distributed into 10 tubes per replicate and per cell line (n = 2 biologically independent samples).

For all TPP experiments, the heating step was performed in a SimpliAmpTM Thermal Cycler (Thermo Fischer Scientific) for 3 min at the following temperatures: 37, 41, 44, 47, 50, 53, 56, 59, 63, and 67 °C. Then the samples were left 3 min at RT. Samples for the lysate experiment were transferred to thick-wall polycarbonate ultracentrifuge tubes (Beckman-Coulter) at this point and samples for cell experiment were snap frozen in liquid nitrogen for cell lysis. The lysis was performed using five freezing/thawing steps with vortexing after each thawing. Finally, all lysates were transferred to ultracentrifuge tubes, placed into a Ti 42.2 rotor (Beckman-Coulter) and ultracentrifuged at 100,000 × g for 20 min using an Optima XPN-80 Ultracentrifuge (Beckman-Coulter). Seventy microliters of the supernatant was collected and the same volume of lysis buffer (8 M urea, 50 mM Tris pH 8.5) was added and the protein concentration was measured using Pierce bicinchoninic acid assay (BCA) protein assay kit (Thermo Fischer Scientific) according to the manufacturer's protocol.

**Microarray RNA analysis**. The RNA isolation had been performed using the miRNeasy mini kit (Qiagen) according to the manufacturer's protocol with DNaseI (Qiagen) treatment. The RNA integrity number (RIN) values were higher than 9.5 for all RNA samples. The microarray analysis was performed according to the manufacturer's instructions (Termo Fischer Scientific UserGuide P/N 703174). Procedures for cDNA synthesis, target DNA fragmentation, and labelling were carried out according to the GeneChip WT PLUS Reagent Kit (Applied Biosystems) using 500 ng of total RNA as the starting material. Hybridization on Affymetrix Gene Chip Human Transcriptome Array 2.0 microarrays, array washing, staining, and scanning were performed according to the manufacturer protocol. GeneChip Scanner 3000 7 G system (Affymetrix) was used to scan the microarrays and the scans were converted into CEL files using the scanner software and were then processed using the Transcriptome Analysis Console 3.0.

**Expression proteomics experiments**. EBs were induced as described above using both iPSCs and ESCs and differentiated for 9 days. iPSCs, hFF, RKO, HT29, and the human neurons differentiated from iPSCs were grown as described in the cell culture section. Then cells were rinsed two time with PBS and lysed in the plate with 1% SDS, 8 M urea, 50 mM Tris buffer pH 8.5. The cells were then sonicated using Branson probe sonicator for 45 s with a 3 s pulse at 30% amplitude. The protein concentration was measured by BCA assay followed by sample preparation for proteomics analysis.

**Density gradient analysis**. Sucrose density gradients were prepared prior to sample preparation on the day of the experiment and stored at 4 °C. Two sucrose solutions were prepared, corresponding to 10 and 50% sucrose concentration dissolved in 20 mM HEPES pH 7.6, 100 mM KCl, 5 mM MgCl$_2$, 10 μg/mL cycloheximide, and 10 U/mL RNase inhibitor and protease inhibitor cocktail (Thermo Fischer Scientific). Ten to fifty percent sucrose gradients were prepared using a Gradient Master 108 (Biocomp) according to manufacturer's instructions in 13.2 mL polypropylene ultracentrifuge tubes (Beckman-Coulter).

iPSCs and RKO cells were grown until 80% confluence in their respective media. Then the cells were incubated with 100 ug/mL cycloheximide in their respective medium for 10 min at 37 °C. The medium was discarded, and the cells were transferred on ice and washed two times with ice-cold PBS containing 100 μg/mL cycloheximide. Cells were then scraped in PBS with cycloheximide and pelleted at 340 × g at 4 °C in a 5804 R centrifuge (Eppendorf) for 3 min. PBS was discarded and the pellets were resuspended in lysis buffer composed of 20 mM Tris-Cl pH 7.4, 150 mM NaCl, 5 mM MgCl$_2$, 1 mM DTT, 1% Triton-X100, 100 μg/mL cycloheximide, 25 U/mL of RNase inhibitors (Thermo Fischer Scientific) and protease inhibitors. Cells were then vortexed for 15 s and centrifuged at 20,000 × g for 6 min. The supernatant was transferred to a new tube and RNA levels were measured with a DS-11 Spectrophotometer (DeNovix). Then 500 μL of the cell lysate corresponding to 6 OD was added on top of the sucrose density gradient after removing 500 μL from the top of the gradient.

The gradients were centrifuged in a SW 41 swinging bucket rotor (Beckman-Coulter) in an Optima XPN-80 Ultracentrifuge at 36,000 rpm for 2 h at 4 °C with maximum acceleration and no brake option. After centrifugation, the gradients were placed at 4 °C and analyzed one by one using BR-188 Density Gradient Fractionation System (Brandel). The parameters were 1.5 mL/min speed and 1.0

abs of resolution. The chase solution consisted of 60% sucrose. Fractions were collected every 30 s until the end of the gradient and absorbance at 245 nm was monitored using Brandel Peak Chart recording software version 2.08.

For iPSC and RKO, fractions corresponding to the soluble part of the gradient, 40S + 60S, 80S mRNA with 2 and 3 bound ribosomes (light polysome) and mRNA with 4 and more bound ribosomes (heavy polysome) were selected (Fig. 4a). Proteins were precipitated using methanol chloroform followed by sample preparation for proteomics analysis.

**SBDS siRNA treatment of hFF**. hFF were seeded in IMDM medium supplemented with 10% FBS at a density of 30,000 per well in six-well plates. 24 h later, they were transfected with either SBDS or scrambled siRNA (Qiagen) using lipofectamine 2000 (Thermo Fischer Scientific) according to the manufacturer's instructions. All siRNA and primers used in this study are presented in Supplementary Table 1. Briefly, for each reaction 5 μL of lipofectamine were added to 240 μL of IMDM medium without serum and incubated at RT for 5 min. Then, 5 μL of 20 μM solution of either SBDS or AllStars negative control siRNA (Qiagen) (Scrambled siRNA 1) were added, and the mix was incubated at RT for 20 min. SBDS siRNA correspond to Hs_SBDS_5 FlexiTube siRNA (SBDS siRNA 1) from Qiagen, which is a functionally verified siRNA targeting 5′-TTGGAAGTACTCA ATCTGAAA-3′ sequence with 5′-GGAAGUACUCAAUCUGAAATT-3′ (sense) and 5′-UUUCAGAUUGAGUACUUCCAA-3′ (antisense) siRNA duplex sequences. hFF medium was replaced by 1 mL of fresh medium and the respective lipofectamine/siRNA reactions were added drop by drop on top of the hFF. Cells were incubated overnight with the lipofectamine/siRNA mix and then the medium was changed for fresh medium. After 2- and 4-days triplicates of SBDS siRNA and AllStars negative control siRNA treated hFF were washed two times with PBS and lysed using 1% SDS, 8 M urea, 50 mM Tris pH 8.5. Finally, the cells were scraped and sonicated as described in the Expression proteomics section, followed by sample preparation for proteomics analysis.

**SBDS KD in EBs**. EBs were differentiated from hi12 as described in Fig. 7a. Three days after plating of EBs on gelatin (6th day of the differentiation), they were treated with either SBDS or AllStars (scrambled) siRNA following the same procedure as for hFFs except that the medium used was MEM. Three days later (9th day of the EBs differentiation), cells were rinsed two times with PBS to prepare for protein and RNA extraction. For proteomics analysis, cells were lysed using 1% SDS, 8 M urea, 50 mM Tris pH 8.5. Later, cells were harvested and sonicated as described in the Expression proteomics section, followed by sample preparation for proteomics analysis. For qRT-PCR, samples were prepared as described below.

**SBDS KD in PSCs transferred on Geltrex**. hi12 and H9 were maintained as described above, when reaching full confluence cells were detached and transferred to dish coated with Geltrex™ LDEV-Free, hESC-Qualified, Reduced Growth Factor Basement Membrane Matrix (Gibco™) at 1/20 dilution in Nutristem medium. Two days after plating, cells were treated with the two siRNAs (Scrambled siRNA 1 and SBDS siRNA 1) as described above. One new siRNA targeting *SBDS* mRNA, and one new control were also used. The control was *Silencer*™ Select Negative Control No. 1 siRNA (Scrambled siRNA 2) (cat#4390843, Ambion™) and the siRNA targeting SBDS was Silencer® Select siRNA number s27482 (Ambion™)(SBDS siRNA 2), which is a functionally verified siRNA targeting 5′-CGAAAUCGCCUGCUACAA A-3′ sequence with 5′-CGAAAUCGCCUGCUACAAATT -3′ (sense) and 5′-UUU GUAGCAGGCGAUUUCGAA-3′ (antisense) siRNA duplex sequences. Cells were passaged when reaching confluence and another round of treatment was performed for 3 days. After this, cells were rinsed two times with PBS to prepare for protein and RNA extraction and qRT-PCR analysis as described below.

**SBDS protein knockin in hi12 and H9**. hi12 and H9 were grown as described above until around 80% confluence, then they were treated with lipofectamine containing the plasmid with SBDS gene or empty as a control. For each reaction 2 μg of plasmid were added to 4 μl of lipofectamine 2000 in 200 μl DMEM medium, which was then added on top of the cells. The plasmid characteristic was as follow: Vector: pcDNA3.1/Zeo (−), Ampicillin, 5014 bp (GeneScript). The SBDS gene sequence was inserted in 5′ Default Cloning Site: XhoI CTCGAG. 5′ start codon: ATG, 3′ stop codon: TGA, 3′ default cloning site: BamHI GGATCC. The clone ID is OHu18189C, containing the following sequence: ORF Clones (Accession No.): NM_016038.3 (ORF Sequence) and the final sequence of the inserted gene is: CTCGAGATGTCGATCTTCACCCCCACCAACCAGATCCGCCTAACCAATGT GGCCGTGGTACGGATGAAGCGTGCCGGGAAGCGCTTCGAAATCGCCTGC TACAAAAACAAGGTCGTCGGCTGGCGGAGCGGCGTGGAAAAAGACCTC GATGAAGTTCTGCAGACCCACTCAGTGTTTGTAAATGTTTCTAAAGGTCA GGTTGCCAAAAAGGAAGATCTCATCAGTGCGTTTGGAACAGATGACCAA ACTGAAATCTGTAAGCAGATTTTGACTAAAGGAGAAGTTCAAGTATCAG ATAAAGAAAGACACACACAACTGGAGCAGATGTTTAGGGACATTGCAAC TATTGTGGCAGACAAATGTGTGAATCCTGAAACAAAGAGACCATACACC GTGATCCTTATTGAGAGAGCCATGAAGGACATCCACTATTCGGTGAAAA CCAACAAGAGTACAAAACAGCAGGCTTTGGAAGTGATAAAGCAGTTAAA AGAGAAAATGAAGATAGAACGTGCTCACATGAGGCTTCGGTTCATCCT TCCAGTCAATGAAGGCAAGAAGCTGAAAGAAAAGCTCAAGCCACTGATC

AAGGTCATAGAAAGTGAAGATTATGGCCAACAGTTAGAAATCGTATGT CTGATTGACCCGGGCTGCTTCCGAGAAATTGATGAGCTAATAAAAAAGG AAACTAAAGGCAAAGGTTCTTTGGAAGTACTCAATCTGAAAGATGTAGA AGAAGGAGATGAGAAATTTGAATGAGGATCC. The day after transfection, the medium was replaced with fresh Nutristem medium containing 5 μg/ml Zeocin™ Selection Reagent (Gibco™). After 1 day of treatment cells were rinsed two times with PBS to prepare for protein and RNA extraction and qRT-PCR analysis as described below.

Since there was no increase in expression of SBDS in hi12 after transfection with lipofectamines, we opted for transfection using electroporation. hi12 at full confluence were detached and $3 \times 10^6$ cells per sample were aliquoted, prepared for electroporation using P3 Primary Cell 4D-NucleofectorTM X Kit L (Lonza) according to manufacturer's protocol, with 5 μg of the same pcDNA3.1/Zeo (−) plasmid as above, containing SBDS gene or empty, and cells were electroporated in a 4D-Nucleofector® (Lonza) using a CT150 pulse. After the pulse, cells were seeded on laminin-521-coated plates in Nutristem medium. After 48 h of culture, cells were treated with 5 μg/ml Zeocin for an additional 48 h and then rinsed two times with PBS to prepare for protein and RNA extraction and qRT-PCR analysis as described below.

**Quantitative real-time PCR (qRT-PCR)**. Total RNA was isolated using RNAeasy Microprep kit (QIAGEN) according to the manufacturer's instructions. cDNA was synthesized with 0.2 μg of total RNA in 20 μL reaction mixture using High Capacity RNA-to-cDNA kit (Thermo Fischer Scientific) according to the manufacturer's instructions. Real-time quantitative RT-PCR Taqman assays were performed using CFX Connect™ Real-Time PCR Detection System (Bio-Rad). All reactions were done in quadruplicates with the use of a pre-developed gene expression assay mix (Applied Biosystems) containing primers and a probe for the messenger RNA of interest. Additionally, each experiment included the assay mix for *GAPDH* for normalization of the RNA input. All data were analyzed using CFX manager version 3.0 (Bio-Rad). All the signals appearing at the 35th and later cycles were regarded as not detected.

**Protein sample preparation for expression proteomics and TMT10 and TMT11 labeling**. For all proteomics experiments, 50 μg of proteins were used in sample preparation. Reduction was performed using 5 mM DTT at RT for 1 h followed by alkylation using 15 mM IAA at RT in the dark for 1 h. The reaction was quenched by adding 10 mM DTT. Then methanol/chloroform precipitation was performed as followed: 3 sample volume of methanol were added, then 1 sample volume of chloroform and 3 volumes of water. Samples were vortexed between each step and then centrifuged at $20,000 \times g$ for 10 min at 4 °C. The aqueous layer was removed, and the protein pellet was rinsed with one sample volume of methanol, vortexed, and centrifuged using the same speed as in the previous step. Finally, all the liquid was removed, and the protein pellet was air-dried.

Air-dried protein pellets were resuspended in 8 M urea, 20 mM EPPS pH 8.5. The samples were diluted once with by adding 20 mM EPPS pH 8.5 (4 M urea) and lysyl endopeptidase digestion was carried out at a 1:100 ratio (LysC/protein, w/w) overnight at RT. The following day, samples were diluted four times (1 M urea) with 20 mM EPPS pH 8.5, then tryptic digestion was performed for 6 h at RT using a 1:100 ratio (Trypsin/protein, w/w). For PISA, a sample composed of one tenth of each sample pooled together was prepared as technical replicate for normalization purposes. After that, TMT10 and TMT11 labeling were performed during 2 h at RT by adding 0.2 mg of reagent dissolved in dry ACN according to manufacturer's instructions and a final ACN concentration of 20%. The reaction was then quenched by adding triethylamine to a final 0.5% concentration and incubated 15 min at RT. The samples were combined resulting in one pooled sample per replicate containing each temperature. After that, the samples were acidified to pH < 3 using TFA, desalted using Sep Pack (Waters) and vacuum dried overnight using miVac DNA (Genevac).

**High-pH reversed-phase peptide fractionation**. For experiments 2–5 and 10 (Supplementary Table 2) 70 μg of peptides were resuspended into 300 μL of 0.1% TFA and fractionated using Pierce™ High pH Reversed-Phase Peptide Fractionation Kit (Thermo Fischer Scientific) according to manufacturer's protocol, resulting in 8 fractions per samples. Samples were then dried overnight in a Speedvac.

For experiments 1, 7–9, 11, and 12 (Supplementary Table 2), 150 μg of peptides were resuspended into 20 mM NH₄OH. Then, samples were off-line high-pH reversed-phase fractionated[63,64] using an Ultimate™ 3000 RSLCnano System (Dionex) equipped with a XBridge Peptide BEH 25 cm column of 2.1 mm internal diameter, packed with 3.5 μm C18 beads having 300 Å pores (Waters). The mobile phase consisted of buffer A (20 mM NH4OH) and buffer B (100% ACN). The gradient started from 1% B to 23.5% in 42 min, then to 54% B in 9 min, 63% B in 2 min and stayed at 63% B for 5 min and finally back to 1% B and stayed at 1% B for 7 min. This resulted in 96 fractions that were concatenated into 24 fractions and dried o/n using miVac DNA.

**Mass spectrometry analysis**. Prior to mass spectrometry analysis, all samples were resuspended in 2% ACN and 0.1% FA at a concentration of 0.2 μg/μL and 1 μg was injected into the respective LC system (summarized in Supplementary

Table 2). Mass spectra were acquired using the parameters listed in Supplementary Table 3.

**Oxygen consumption and extracellular acidification**. Metabolic flux analysis was performed on hi12, hFF, and RKO cells using Seahorse XF24 Extracellular Flux Analyzer (Seahorse Biosciences, Billerica, MA). Two days before the experiment 15,000 cells/well were seeded. hi12 were seeded after coating the wells with laminin-521. The day of the experiment the medium was changed in 500 μL DMEM/F12 low buffer capacity containing either 2% FBS (hFF, RKO cells) or 20 μL/mL of E8 supplements (hi12 cells). The cells were incubated at 37 °C without $CO_2$ 1 h prior to experiment.

The oxygen consumption rate (OCR) and acidification rate (ECAR) were recorded at basal level and following injection of oligomycin (1 μM final), carbonyl-cyanide 4-trifluoromethoxy-phenylhydrazone FCCP (1–2 μM) and mixture of rotenone and antimycin A (1 μM). Running template was 2 min mix, 1 min wait and 4 min measure. All chemicals were purchased from Sigma–Aldrich. Data were normalized on the number of cells per well and against basal OCR and ECAR. Normalization for cell number was carried out staining the nuclei in 500 μL of Hoescht 33342 (Molecular Probes) for 10 min and then imaging each well using BD pathway 855 (BD Biosciences, Franklin Lakes, U.S.) with ×10 objective and montage 5 × 4. Cell numbers were counted with Cell profiler software.

**Protein synthesis rate measurement in hi12, hFF, and RKO**. Protein synthesis rate in hi12, hFF, and RKO was measured using Click-iT® Plus OPP Protein Synthesis Assay (Thermo Fischer Scientific). Briefly hi12, hFF, and RKO were treated with OPPuro at exponential growth phase and samples were processed according to manufacturer's protocol and prepared for flow cytometry analysis. Single-cell suspension was prepared by filtering through 40 μm cell strainers (BD Falcon). NuclearMask™ Blue stain was used to gate on cells, after which mean fluorescence intensity (MFI) of the FITC channel was calculated. A SORP BD LSRII Analytic Flow Cytometer (BD Biosciences) was used for acquisition and the data was analyzed with Flowjo 8.8.6 (Tree Star Inc., OP, USA). P-values were calculated using a two-sided Student $t$-test.

**Protein synthesis rate measurement in SBDS siRNA treated hFF**. 3000 hFF were seeded in 96-well plates and the day after they were treated with SBDS siRNA and AllStars negative control siRNA following the same procedure as described above for 3 days ($n = 6$ biologically independent samples for SBDS and control). Then protein synthesis rate was measured using OPPuro according to manufacturer's protocol and analyzed using Operetta.

**Quantification and statistical analysis**
*TMT10 labelling quantification*. Protein identification and quantification were performed using MaxQuant software (version 1.6.2.3). MS2 was selected as the quantification mode with TMT10 or TMT11 as the modification. Acetylation of N-terminal, oxidation of methionine and deamidation of asparagine and glutamine were selected as variable modifications. Carbamidomethylation of the cysteine was selected as fixed modification. The Andromeda search engine was using the Uni-Prot human database 2019_05 excluding protein isoforms (73 910 entries) with the precursor mass tolerance for the first searches and the main search set to 20 and 4.5 ppm, respectively. Trypsin was selected as the enzyme, with up to two missed cleavages allowed; the peptide minimal length was set to seven amino acids. Default parameters were used for the instrument setting. The FDR was set to 0.01 for peptides and proteins. "Match between runs" option was used with a time window of 0.7 min and an alignment time window of 20 min.

*Analysis of microarray mRNA data*. Statistical analysis of differential gene expression was performed using a nonpaired one-way (ANOVA) test with the Benjamini–Hochberg correction for multiple comparisons. For the detection of differentially expressed genes, the threshold for the adjusted $p$-value was set at 0.01, and the threshold for the fold change (FC) was set at 2.0. Probe sets with no associated Gene Symbol were excluded from the analyzes.

*Data normalization and statistical analysis*. All data analysis and plots were produced using R version 3.6.1 and Excel 2016. In all proteomics analysis, individual protein abundances (corrected reporter ion intensity of LFQ intensity) were normalized by the sum of all protein abundances in the corresponding sample. Analysis of TPP data was performed using TPP R package version 3.13 (Bioconductor) according to the procedure described by Franken et al.[10] Estimated protein melting points (Tm) were extracted from the result file produced by the TPP package and used for data analysis (Supplementary Data 3). Comparison between two sample groups were assessed using two-tailed unpaired $t$-test unless otherwise specified. Data are presented as mean ± standard error of the mean, unless otherwise stated. P-values lower than 0.05 were considered to be statistically significant. $r$ for the corresponding plots were calculated using Pearson correlation. For the analysis of the data from metabolic flux analysis, a Grubb's test was used to remove outliers among biological replicates. GO pathways enrichment were done using Gene Ontology enRIchment anaLysis and visuaLizAtion tool (Gorilla)[65,66] or DAVID version 6.8. For pathways enrichment significance, we considered

pathways with $p$-values corrected by Benjamini–Hochberg procedure lower than 0.05. Only proteins identified with two peptides or more and without missing values in any of the samples and replicates were considered for statistical analysis.

**Reporting summary**. Further information on research design is available in the Nature Research Reporting Summary linked to this article.

## Data availability
The authors declare that all data supporting the findings of this study are available within the paper and its supplementary information files. All relevant data are available from the corresponding authors (R.A.Z. and S.R.). The mass spectrometry proteomics data have been deposited to ProteomeXchange Consortium (http://proteomecentral.proteom exchange.org) via the PRIDE partner repository with dataset identifier PXD018453 (PISA-Express data), PXD014830 (TPP data of hi11, hi12, hi13, hFF, RKO in cells; TPP data of hi12 and hFF in cell lysates; ribosomal protein expression data in hi12 and RKO; protein expression data in hFF, hi12, HT29, Neurons, EBs, RKO; SBDS KD protein expression data in hFF; SBDS KD protein expression data after EB induction in hi12) and PXD015874 (Protein expression data of EB induction in H9 and HS980). The microarray RNA analysis data were deposited to GEO repository under the accession number GSE135409. Statistics from the TPP data generated by the TPP R package version 3.13 from Franken et al.[10] are deposited in the Zenodo repository at https://doi.org/10.5281/ zenodo.5018241. The ProteoTracker tool is freely available at http://www.proteotracker. genexplain.com. Source data are provided with this paper.

## Code availability
The code for the web-interface is available in GitHub (https://github.com/RZlab/ ProteoTracker) and Zenodo (https://doi.org/10.5281/zenodo.5549677)[67].

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

## Acknowledgements

This study was supported by the Knut and Alice Wallenberg Foundation (grant KAW 2014.071 to R.A.Z. and 2015.0063 to R.A.Z. and R.H.); Uppsala County Council (to K.H.G.); RuFu (to K.H.G.); the Swedish Foundation for Strategic Research and the Swedish Government, StemTherapy (to K.H.G. and to S.R.); the HSE University Basic Research Program (to D.M.); the Russian Science Foundation project # 17-14-01338 (to A.G.T.); Novo Nordisk A/S, SRP Diabetes; the Swedish Research Council and the Family Erling-Persson Foundation. A.A.S. was supported by Swedish Research Council (grant 2020-00687) and the Swedish Society of Medicine (grant SLS-961262, 1086 Stiftelsen Albert Nilssons forskningsfond). R.A.Z. acknowledges the Ministry of Science and Higher Education of the Russian Federation (agreement no. 075-15-2020-899).

## Author contributions

S.R. and P.S. conceived the study, performed most of the experiments, analyzed the data, and wrote the manuscript. A.A.S. and N.L. helped with TPP analysis. C.M.B. and A.A.S. helped with statistical and bioinformatics analysis. C.M.B. created the R shiny package. M.A. and R.H. performed the protein synthesis rate measurements and data analysis. N.M. and P.-O.B. performed the metabolic flux and data analysis. D.M. and A.G.T. performed the microarray analysis and statistical analysis. M.N., P.M., S.R., J.C.V., O.S., S.K., and K.-H.-G. repro-grammed iPSCs and characterized them for pluripotency. V.M. and J.C.V. made neurons and characterized them. K.A. and A.K. provided the infrastructure and contributed to the implementation of the web-interface online. M.G. helped with the SBDS knockdown analysis. A.C. performed the qRT-PCRs and analyzed the data. R.A.Z. conceived the study, directed experiments, analyzed the data, and wrote the manuscript.

## Funding

## Competing interests

The authors declare no competing interests.
