## [Peer Review File · Nature Communications]

REVIEWER COMMENTS

Reviewer #1 (Remarks to the Author):

Protein thermal stability changes can be induced by alterations in post-translational modification, composition or structure of protein complexes, metabolite binding, etc. Compared with the conventional investigation at protein abundance level, monitoring protein thermal stability changes in a biological process can add another layer of information from an orthogonal perspective. Former papers have also proved that protein thermal stability does change under certain conditions.

Sabatier et al. designed a method that can measure protein abundance and protein thermal stability differences at the same time. The method was applied to study the transition between human pluripotent stem cells and differentiated cells. Follow-up data analysis and experiments revealed a protein (SBDS) that regulates translation during stem cells maintenance and differentiation. The method also resolves the problem how to compare thermal stability when protein abundance varies among samples. This can be adopted in other biological topics of interest and answer more interesting questions.

The manuscript was well written and provided sufficient details. My major concern is about the validation for SBDS. Overall the results for SBDS are weak. More experiments and data should be provided in order to claim SBDS is a key regulator of translation during stem cells maintenance and differentiation, which is stressed by the title. Some statistical cutoffs also need justifications. Overall the method is definitely of interest to the field, but the validation part must be strengthened before it gets accepted.

Major concerns:

1. Only a loose p-val cutoff is used in Fig. 2e (nominal p 0.05). Considering the sample size is not big, this criteria is concerning. Strict cutoffs or better statistical strategies should be adopted.
2. Page 8, line 179, "As expected, there was no correlation between the FCs in stability versus expression for individual proteins, confirming that changes in these two analyses dimensions occur independently of each other during cell-type transitions". Is there a figure missing? Which figure shows "no correlation between the FCs in stability versus expression"?
3. Page 13, line 292, "It was also unlikely to be caused by a difference in PTMs, as the latter would reduce the level of unmodified peptides, resulting in altered expression read-outs". This is not quite precise. Depending on how peptide quant info is collapsed into protein quant information, and depending on how many peptides are modified, difference in PTMs not always results in altered expression read-outs.

4. Fig 6b and Fig6c, what's the sample size? The effect size is so small, error bars are big. It's hard to draw the conclusion that "SBDS expression anticorrelated with...". Especially fig 6c, MRPs and MRBPs are not changing over time, it's improper to claim "anticorrelated".
5. Fig 6g, what are the p values? It doesn't look like NANOG or OCT4 is significant.
6. There are many barplots in Fig6 and other figures. Sample sizes should be indicated in the figures. And if sample size is 3-5, individual dots should displayed on top of the barplots as well.
7. In validation experiments, many effect sizes are very small. Sample size is not big either. And only one siRNA targeting SBDS and one negative control siRNA were used. At least one more siRNAs and one more negative control siRNA should be included to confirm the small changes. An overexpression experiment or rescue experiment should also be conducted to confirm the phenotype.
8. The p value and fold change cutoffs for Fig 6h are loose. Please justify them, or do something like a permutation test to provide an estimated FDR.

Minor points:

1. Page 3, line 51, "Strikingly, pluripotent cells featured destabilized ribosomal proteins and consistently reduced expression in only one ribosome maturation factor, Shwachman-Bodian-Diamond Syndrome protein (SBDS)". The authors captured only one ribosome maturation factor in the experiment. But it does mean only one factor shows reduced expression in the process. There could be more and they were just missed/not identified by MS. This sentence should be rewritten to be precise.
2. Fig. 2c and Fig. S4b can be combined into a new Fig. 2c., just add the number of proteins with $FC > 3sd$ in Fig. S4b and the new figure will be more informative than two separate ones.
3. Page 9, line 193, add references for Fischer method.
4. Several GO enriched entries in Fig. 2e were described in the text in page 9. It helps if these entries are coded in a different color in the figure.
5. Fig 6b, Fig 6c, the dots can be jittered so that error bars don't overlap.
6. Page 32, line 744 and line 748, it should be "peptides", not "protein".
7. Are total reporter ion intensity of each channel normalized to be equal? Please add the detail in the method.
8. Page 35, line 828, it should be "r", not "r2", as there are negative values in Fig. S4d.
9. What are the p value (or adjusted p value) cutoff and fold enrichment cutoff used in the GO analysis?

10. The S_m can be further elaborated. For example, something like “higher S_m indicates greater thermal stability”

11. Scale bars are missing in Fig S7.

Suppl. tables:

1. Raw results (e.g., reporter ion intensity) should be provided, not just the ratios vs linker. It will facilitate the re-analysis of the data by the community.

2. Table S2-1, what are those mean values? They are tiny. Are they ratios vs the linker? I guess not. They are just so tiny and if they are ratios vs the linker, these tiny values could compromise the reliability of the quantification.

3. Table S2-3, 2-4, 2-5, what are those normalized expression values? They are also super tiny.

4. Table S3, some QC statistics (e.g. R2 and SSE, there may be more) should be appended to the results from TPP package.

5. Table S4, “same than above” should be “same as above”. There is also a typo “Fischer” in the table.

Reviewer #2 (Remarks to the Author):

In this manuscript Sabatier et al. combined expression proteomics with the proteome-wide integral solubility alteration (PISA) assay to quantify protein abundance and protein solubility/thermal stability in different human cells including developmentally related cells such as iPSCs, ESC, and embryoid bodies and the colon cancer cell line PKO. The results obtained from the analysis has been uploaded and visualized in a web-based interface, ProteoTracker. Comparing undifferentiated cells with differentiated cells they concluded the change in proteins involved in metabolic processes precedes the change in components of chromatin remodeling complexes during differentiation. They also found that the quantity of functional ribosomes is lower in undifferentiated cells. Also, ribosomal proteins and the factors involved in ribosome maturation (e.g. SBDS) undergo the most dramatic change in level and stability during differentiation.

Major Comments:

1-The study generated a comprehensive proteomic database that can have long-lasting impact on the field of embryonic stem cells and differentiation. Particularly the study of protein thermal stability in undifferentiated iPSC/ESCs and after differentiation is novel and interesting. However, it is unclear (at least to this reviewer) what is the significance of thermal stability in overall protein function. No experiment has been conducted to examine the effect of change in thermal stability on stem cells or differentiation.

2-The Authors refer to Microarray analysis, but the relevant data do not seem to appear in the figures, as I could not find them. Lack of transcriptomic data leaves the question open as to whether the changes in protein levels stems from the change in transcription or translation or stability or other post-transcriptional events.

3-The presentation of the data in the current format is not very informative. Panels that only describe the methods or give a general summary of data without giving detailed information about the identified targets such as Figures 1, 2, and 3 and panel 6A should be transferred to supplementary, unless they are acutely necessary to support the main message of the paper.

Minor Comments:

4-In several instances the authors refer to the colon cancer cell line PKO as an example of differentiated cells, which is not correct. In several cellular processes (e.g. cell cycle, metabolism and expression of pluripotency factors) cancer cells are in fact more similar to undifferentiated cells than differentiated cells. For example in Figure 4 it is more relevant to compare iPSCs with ESCs and EBs than RKO.

5-SBDS KD cell should be examined by polysome profiling to assess the impact of SBDS on ribosome maturation and 60S peak.

Reviewer #3 (Remarks to the Author):

I understand why there is an interest in cell fate transitions between pluripotent stem cells and their derivatives. But the choice of cell types used here is not well introduced or justified, and the term 'differentiated' cell types is so broad that it lacks context. Pluripotent stem cells can give rise to cells from each germ layer (endoderm, ectoderm, and mesoderm), so it seems to me that a more compelling study would have used cells from various developmental stages within a single lineage to track transitions, or examined comparable states across lineages. Here, the cell types compared include pluripotent, embryoid body (a heterogeneous mixture of cell types from each germ layer that has already undergone many cell transitions between pluripotent and formation of EB), foreskin fibroblasts (terminally differentiated and of mesoderm origin), and stem cell derived neuronal cells (intermediately differentiated ectoderm derived). Also included is a cancer cell line that is derived from epithelial cells likely of endoderm origin that otherwise has no clear biological relevance for comparison to pluripotent stem cells other than having a fast growth rate. A comparison of EB to pluripotency is more than just a difference in stemness, as is differences between fibroblasts and EB. However, to package this as a story focused on stemness and cell transitions does not seem well grounded as currently presented. Adding to this, there is no inclusion of varied cell culture conditions which are known to affect stemness and proliferation rates, including passage number, cell density, use of chemically defined media vs use of FBS, substrate, etc. What was the passage number of each pluripotent stem cell culture used for MS? It appears that only a single culture condition was used for each cell type. Thus, at best, these are comparisons of static and distinct cellular phenotypes for which the differences can be attributable to contributions from cell fate, growth conditions, proliferation rates, etc. Statements like "insights into molecular bases of differentiation processes" are therefore an over-interpretation of the data and figure 2d is misleading as it gives the impression that the cell types used here are all within a single lineage. This is not to say that the body of work is not substantial. There is obviously a tremendous amount of work described and the technical aspects appear sound. The methodology described will also likely be useful for other studies. The idea of ribosome biogenesis emerging from this proteomics method is also intriguing. Therefore, it seems prudent to reformulate the message of the paper.

ProteoTracker is described as comprising a plurifaceted dataset and online software for comprehensive system-wide analysis of thermal stability and expression level of proteins. Based on this description of it being a comprehensive tool, I expected to be able to upload my own data. However, the web tool appears to only show data from this study. So, in that regard it is less of an online software tool for comprehensive system-wide analysis, and more of a website for displaying or visualizing data presented in the current study – essentially it is an interactive supplementary data presentation. This also lessens the interest in the tool / data for other labs that study pluripotency and early differentiation, as the biological significance of the cell types used here are limited, as described above. The website would benefit from a user guide and detailed descriptions or figure legends so that the graphs that are generated can be interpreted appropriately. Plots that can be downloaded as SVG would be more useful than PDF.

In the manuscript figures, bar graphs are not appropriate for displaying quantitative data. Rather, dot blots or similar format which displays datapoints should be used.

Outline of the replies to Reviewers' comments. They are marked X.Y, where X is the Reviewer #, and Y is the issue number for a given Reviewer. If the reviewer's comment raised more than one issue, each one was considered and numerated separately. The original numbers given by the Reviewer are provided in parentheses.

REVIEWER COMMENTS

Reviewer #1 (Remarks to the Author):

Major concerns:

1.1 (1). Only a loose p-val cutoff is used in Fig. 2e (nominal p 0.05). Considering the sample size is not big, this criteria is concerning. Strict cutoffs or better statistical strategies should be adopted.

Reply: A relatively loose cut-off in Fig. 2e was employed on purpose, as the selected proteins passed through the additional filter, Gene Ontology classification. The sufficiency of the chosen statistical threshold is supported by the high enrichment of the GO terms for many transitions: e.g., for B->A, the enrichment was 20-40 fold. In addition, the Fischer score cutoff can be adjusted manually on the web interface to fit the user's preferences.

1.2 (2). Page 8, line 179, "As expected, there was no correlation between the FCs in stability versus expression for individual proteins, confirming that changes in these two analyses dimensions occur independently of each other during cell-type transitions". Is there a figure missing? Which figure shows "no correlation between the FCs in stability versus expression"?

Reply: We have added Extended Data Fig. 4b that includes the Pearson correlation score between the fold change in stability versus expression in each cell line against iPSC.

1.3 (3). Page 13, line 292, "It was also unlikely to be caused by a difference in PTMs, as the latter would reduce the level of unmodified peptides, resulting in altered expression read-outs". This is not quite precise. Depending on how peptide quant info is collapsed into protein quant information, and depending on how many peptides are modified, difference in PTMs not always results in altered expression read-outs.

Reply: We agree. The text is corrected to "There are doubts that it was caused by a difference in PTMs, as the latter would reduce the level of unmodified peptides, which may result in altered expression read-outs".

1.4 (4). Fig 6b and Fig6c, what's the sample size?

Reply: The sample size is 3 in both Figs. 6b and 6c but the measurements in Fig. 6c encompass the mean expression of 200+ proteins that are annotated to play a role in ribosome biogenesis and ribosome maturation according to the Uniprot database.

1.5 (5). The effect size is so small, error bars are big. It's hard to draw the conclusion that "SBDS expression anticorrelated with...". Especially fig 6c, MRPs and MRBPs are not changing over time, it's improper to claim "anticorrelated".

Reply: We agree that anticorrelation may be too strong a statement. The text was modified to "opposite trend" or "opposite protein expression behavior".

1.6 (5). Fig 6g, what are the p values? It doesn't look like NANOG or OCT4 is significant.

Reply: Fig. 6 is modified, panels 6e and 6f were moved to Extended Data Fig. 8a and 8b, panel 6h is now in Fig.7a. Panel 6g is now panel 6e and we added the p-values in the Figure, both NANOG and OCT4 are significant using student *t*-test.

1.7 (6). There are many barplots in Fig6 and other figures. Sample sizes should be indicated in the figures. And if sample size is 3-5, individual dots should displayed on top of the barplots as well.

Reply: we changed all the main and supplementary figures accordingly.

1.8 (7). In validation experiments, many effect sizes are very small. Sample size is not big either. And only one siRNA targeting SBDS and one negative control siRNA were used. At least one more siRNAs and one more negative control siRNA should be included to confirm the small changes.

Reply: We performed the experiment with one new siRNA targeting SBDS and one more control. The results show that OCT4 mRNA levels increase significantly upon SBDS KD and several differentiated lineage markers were decreased in iPSCs. In ESCs, the expression of OCT4 and NANOG increased significantly, and the expression of several differentiation markers was decreased. These results are compiled in a new Fig. 7 panels a, b and c. The experiments are described in the method section.

1.9 (7), An overexpression experiment or rescue experiment should also be conducted to confirm the phenotype.

Reply: We purchased a pcDNA3.1/Zeo (-) (GeneScript) plasmid with Zeocin as selective compound either containing SBDS gene or empty (control). The knock in was performed initially using Lipofectamine 2000 in ESCs and iPSCs. However, the knock in via Lipofectamine delivery significantly increased expression of SBDS only in ESCs despite of several attempts in iPSCs hi12. Then we used electroporation to transfect our iPSCs and managed to significantly increase SBDS expression. The results compiled in Fig. 7d and 7e show that SBDS overexpression led to a significant decrease in expression of OCT4 and NANOG in ESCs and OCT4 in iPSCs. The experiments' details are described in the Methods section.

1.10 (8). The p value and fold change cutoffs for Fig 6h are loose. Please justify them, or do something like a permutation test to provide an estimated FDR.

Reply: Fig. 6h is now Fig. 6f. In permutations of the replicates, only 2 proteins passed our cutoff, while in an unpermuted dataset 155 proteins were significant. We added this information in the text.

Minor points:

1.11 (1). Page 3, line 51, “Strikingly, pluripotent cells featured destabilized ribosomal proteins and consistently reduced expression in only one ribosome maturation factor, Shwachman-Bodian-Diamond Syndrome protein (SBDS)”. The authors captured only one ribosome maturation factor in the experiment. But it does mean only one factor shows reduced expression in the process. There could be more and they were just missed/not identified by MS. This sentence should be rewritten to be precise.

Reply: We clarified the sentence: “Strikingly, PSCs featured destabilized ribosomes. In addition, among the ribosome biogenesis factors that were detected in proteomics experiments, only Shwachman-Bodian-Diamond Syndrome protein (SBDS) showed consistently reduced expression in PSCs.”

1.12 (2). Fig. 2c and Fig. S4b can be combined into a new Fig. 2c., just add the number of proteins with FC>3sd in Fig. S4b and the new figure will be more informative than two separate ones.

Reply: We modified the figures accordingly. A new Figure 2c was created.

1.13 (3). Page 9, line 193, add references for Fischer method.

Reply: We added the references.

1.14 (4). Several GO enriched entries in Fig. 2e were described in the text in page 9. It helps if these entries are coded in a different color in the figure.

Reply: We highlighted the ribosome pathway in bold in Figure 2e since it is the main pathway discussed in the study.

1.15 (5). Fig 6b, Fig 6c, the dots can be jittered so that error bars don't overlap.

Reply: We modified the figures accordingly.

1.16 (6). Page 32, line 744 and line 748, it should be “peptides”, not “protein”.

Reply: We modified the text accordingly.

1.17 (7). Are total reporter ion intensity of each channel normalized to be equal? Please add the detail in the method.

Reply: We normalized the reporter ion intensity by the total sum of all protein intensities for each reporter ion. A sentence has been added in the Methods section.

1.18 (8). Page 35, line 828, it should be “r”, not “r²”, as there are negative values in Fig. S4d.

Reply: We modified the text accordingly.

1.19 (9). What are the p value (or adjusted p value) cutoff and fold enrichment cutoff used in the GO analysis?

Reply: We used a cutoff of 0.05 for p-values corrected by Benjamini-Hochberg procedure to reduce false discovery rate, it is now specified in the Methods section.

1.20 (10). The S_m can be further elaborated. For example, something like “higher S_m indicates greater thermal stability”

Reply: We agree and added a sentence in the text.

1.21 (11). Scale bars are missing in Fig S7.

Reply: We added the missing scale bar.

1.22 (1) Suppl. tables:

1. Raw results (e.g., reporter ion intensity) should be provided, not just the ratios vs linker. It will facilitate the re-analysis of the data by the community.

Reply: We added the raw reporter ion intensities to Extended data tables. We also kept the normalized values as we felt like it would be easier this way for researchers outside the proteomics field to reproduce our results.

1.23 (2). Table S2-1, what are those mean values? They are tiny.

Reply: They are the reporter ion intensities normalized by the summed intensities. We added the raw reporter ion intensities to all Extended data tables.

1.24 (2) Are they ratios vs the linker? I guess not. They are just so tiny and if they are ratios vs the linker, these tiny values could compromise the reliability of the quantification.

Reply: See the answer for point 1.23 above.

1.25 (3). Table S2-3, 2-4, 2-5, what are those normalized expression values? They are also super tiny.

Reply: See the answer for point 1.23 above.

1.26 (4). Table S3, some QC statistics (e.g. R2 and SSE, there may be more) should be appended to the results from TPP package.

Reply: We added the statistics from the TPP package, they are available here:

<https://doi.org/10.5281/zenodo.5018241>

A sentence with the link has been added in the data availability section.

1.26 (5). Table S4, “same than above” should be “same as above”.

Reply: We adjusted the text accordingly.

1.27 (5) There is also a typo “Fischer” in the table.

Reply: We corrected the typo.

Reviewer #2 (Remarks to the Author):

In this manuscript Sabatier et al. combined expression proteomics with the proteome-wide integral solubility alteration (PISA) assay to quantify protein abundance and protein solubility/thermal stability in different human cells including developmentally related cells such as iPSCs, ESC, and embryoid bodies and the colon cancer cell line PKO. The results obtained from the analysis has been uploaded and visualized in a web-based interface, ProteoTracker. Comparing undifferentiated cells with differentiated cells they concluded the change in proteins involved in metabolic processes precedes the change in components of chromatin remodeling complexes during differentiation. They also found that the quantity of functional ribosomes is lower in undifferentiated cells. Also, ribosomal proteins and the factors involved in ribosome maturation (e.g. SBDS) undergo the most dramatic change in level and stability during differentiation.

Major Comments:

2.1-The study generated a comprehensive proteomic database that can have long-lasting impact on the field of embryonic stem cells and differentiation. Particularly the study of protein thermal stability in undifferentiated iPSC/ESCs

and after differentiation is novel and interesting. However, it is unclear (at least to this reviewer) what is the significance of thermal stability in overall protein function. No experiment has been conducted to examine the effect of change in thermal stability on stem cells or differentiation.

Reply: We thank reviewer 2 for such positive comments about our study.

The effect of protein thermal stability on protein function is currently debated in literature¹, but the emerging consensus seems to be that thermal stability changes are as informative as the expression changes^{1,2}. Our data on ribosome stability in PSCs confirms this view. Thermal stability of a protein is affected by numerous factors, including post-translational modifications, protein-protein interactions, ligand binding, association to membrane, changes of compartment, as mentioned in our introduction. Here we demonstrate that for cell type comparison the protein thermal stability measurements offer additional information that would not be accessible with expression measurements alone.

2.2-The Authors refer to Microarray analysis, but the relevant data do not seem to appear in the figures, as I could not find them. Lack of transcriptomic data leaves the question open as to whether the changes in protein levels stems from the change in transcription or translation or stability or other post-transcriptional events.

Reply: We added the data as Extended Data Table 3, they only concern comparison of hFF with iPSCs hi12 and were mainly used to confirm that SBDS is downregulated in iPSCs. We agree with the reviewer that changes in protein expression can stem from various factors, but we are not aiming to investigate all these factors in the present article.

2.3-The presentation of the data in the current format is not very informative. Panels that only describe the methods or give a general summary of data without giving detailed information about the identified targets such as Figures 1, 2, and 3 and panel 6A should be transferred to supplementary, unless they are acutely necessary to support the main message of the paper.

Reply: We believe that these figures are necessary to support the main message of the paper. It is very common to provide method description in Figure 1, because without visualizing the workflow it is hard to understand the results. Figure 2 presents a rather novel and uncommon visualization method in proteomics - Sankey diagram, which therefore needs to be explained to the readers. Figure 3 presents the pathway evolution along the transition and is also important for the main message. Another important part of the article is the cell culture work that contains many experiments at different time points of the differentiation procedure. We feel that it is going to be hard to understand the experiments without a scheme that is presented in Figure 6A and it is a common practice to present differentiation procedures along the pseudo-time axis.

It should be added that multistep cellular transitions is a rather novel item in omics research. As there is no generally accepted way of visualizing them,

especially when more than one parameter is measured along these transitions, we explore here different visualization approaches, providing the opportunity to the reader to choose between them for future use in research.

Minor Comments:

2.11-In several instances the authors refer to the colon cancer cell line PKO as an example of differentiated cells, which is not correct. In several cellular processes (e.g. cell cycle, metabolism and expression of pluripotency factors) cancer cells are in fact more similar to undifferentiated cells than differentiated cells. For example in Figure 4 it is more relevant to compare iPSCs with ESCs and EBs than RKO.

Reply: We agree that some gene expression networks that are responsible for induction and maintenance of pluripotency are also involved in oncogenesis. However, as we demonstrate here RKO has a similar metabolism to hFF (mainly oxidative phosphorylation) rather than to iPSCs (mainly glycolysis), see Extended Data Figures 5c-e. Also, both hFF and RKO do not express pluripotency markers. In Figure 4 we chose RKO to be compared to iPSCs because of the fast division rate. It has been shown that ribosome density profiles may depend on the proliferation rate^{3,4} and, therefore, we selected RKO that has a similar high proliferation rate as iPSCs to show that the difference in ribosomal proteins' stability does not stem from a difference in division rates.

Concerning the comparison of ESCs and iPSCs, we mainly used ESCs as a control for our iPSCs since they should be similar. Indeed, PCA and statistical analyses showed that there are almost no differences between the two cell types in term of protein expression and thermal stability (Figures 2a, 2b and 2c) and that ribosomal proteins change neither expression nor stability between the two cell types (Figure 3c). We also showed later in the confirmation experiments that SBDS behavior during differentiation and upon siRNA KD is similar in iPSCs to that in ESCs.

2.12-SBDS KD cell should be examined by polysome profiling to assess the impact of SBDS on ribosome maturation and 60S peak.

Reply: SDS caused largely by mutations in SBDS gene have been studied and some groups have already shown that SBDS mutation causes a modification of the sucrose density gradient as well as SBDS shRNA KD⁵⁻⁸.

Reviewer #3 (Remarks to the Author):

3.1.1.- I understand why there is an interest in cell fate transitions between pluripotent stem cells and their derivatives. But the choice of cell types used here is not well introduced or justified, and the term 'differentiated' cell types is so broad that it lacks context.

Reply: We agree with reviewer 3 and added a more detailed explanation of the choice of cell lines in the text. We used “differentiated cells” to simplify the text and designate the cell types in the study that are not pluripotent cells, since hFF are not PSC derivatives per se. This is now specified in the Introduction.

3.1.2. Pluripotent stem cells can give rise to cells from each germ layer (endoderm, ectoderm, and mesoderm), so it seems to me that a more compelling study would have used cells from various developmental stages within a single lineage to track transitions, or examined comparable states across lineages.

Reply: We thank reviewer 3 for the highly interesting suggestion. This aspect should be investigated in the future in a separate study.

3.1.3. Here, the cell types compared include pluripotent, embryoid body (a heterogeneous mixture of cell types from each germ layer that has already undergone many cell transitions between pluripotent and formation of EB), foreskin fibroblasts (terminally differentiated and of mesoderm origin), and stem cell derived neuronal cells (intermediately differentiated ectoderm derived). Also included is a cancer cell line that is derived from epithelial cells likely of endoderm origin that otherwise has no clear biological relevance for comparison to pluripotent stem cells other than having a fast growth rate.

Reply: We added a more detailed explanation of the choice of EBs and RKO in this study in the text, with references. EBs formation is one of the most popular way of studying early developmental events in human and is also a precursor for many differentiation protocols, including neuronal, hematopoietic and cardiac lineages⁹⁻¹⁸.

We used RKO to evaluate the magnitude of the differences when comparing cells with the same genome (iPSCs, hFF, and EBs → male) with cells that have a different genome (ESCs → female) and cells that are prone to chromosome rearrangements (RKO → female). This is important since protein thermal stability has only recently started to be applied in cell biology and point mutations are known to affect protein thermal stability¹⁹⁻²². To exemplify the usefulness of this comparison, we surprisingly detected more differences in protein expression between hFF and iPSCs than between RKO and iPSCs, with similar differences in protein thermal stability. We also used RKO to verify that the differences that we observe between PSCs (fast dividing cells) and hFF and EBs (slower dividing cells) do not originate only from a difference in division rate.

A more detailed description is now added in the text.

3.1.4. A comparison of EB to pluripotency is more than just a difference in stemness, as is differences between fibroblasts and EB. However, to package this as a story focused on stemness and cell transitions does not seem well grounded as currently presented.

Reply: We adjusted the text accordingly.

3.1.5. Adding to this, there is no inclusion of varied cell culture conditions which are known to affect stemness and proliferation rates, including passage number, cell density, use of chemically defined media vs use of FBS, substrate, etc.

Reply: We agree with reviewer and have added two sentence (page 10 line 8 and page 19 line 24) that part of the changes may originate from the difference in the cell culture conditions. However, investigation on the influence of cell culture condition on the cellular proteome could represent an additional standalone study.

3.1.6. What was the passage number of each pluripotent stem cell culture used for MS?

Reply: We added passage numbers for the cell lines in the method section.

Concerning the cell density, we started the experiments at around 80% confluence for each cell lines (usually used for exponential growth phase) apart from EBs that expand in all directions when they attach to the culture flask. This is now specified in the Methods section.

3.1.7. It appears that only a single culture condition was used for each cell type. Thus, at best, these are comparisons of static and distinct cellular phenotypes for which the differences can be attributable to contributions from cell fate, growth conditions, proliferation rates, etc. Statements like “insights into molecular bases of differentiation processes” are therefore an over-interpretation of the data and figure 2d is misleading as it gives the impression that the cell types used here are all within a single lineage.

Reply: We agree with reviewer 3 and we modified the text accordingly. For instance:

In the abstract (line 6): “To gain new insight into pluripotency, we applied the method to study transitions between human pluripotent stem cells (PSC) and differentiated cells” was changed to “To gain new insights into regulation of pluripotency, we applied that method to study the differences between human pluripotent stem cells (PSC) and several cell types including their differentiated progeny and their parental cell line”.

Page 5 line 20: “Our data provide novel insights into molecular bases of differentiation processes...” was changed to “This new combined analysis provided a more detailed view of protein behavior after cell type transition...”

The other changes are highlighted in the new revised version of the manuscript.

We also modified figure 2d to avoid confusion.

3.1.8. This is not to say that the body of work is not substantial. There is obviously a tremendous amount of work described and the technical aspects appear sound. The methodology described will also likely be useful for other studies. The idea of ribosome biogenesis emerging from this proteomics method

is also intriguing. Therefore, it seems prudent to reformulate the message of the paper.

Reply: We thank reviewer 3 for these kind words and we modified the text accordingly.

3.2.1.-ProteoTracker is described as comprising a plurifaceted dataset and online software for comprehensive system-wide analysis of thermal stability and expression level of proteins. Based on this description of it being a comprehensive tool, I expected to be able to upload my own data. However, the web tool appears to only show data from this study. So, in that regard it is less of an online software tool for comprehensive system-wide analysis, and more of a website for displaying or visualizing data presented in the current study – essentially it is an interactive supplementary data presentation. This also lessens the interest in the tool / data for other labs that study pluripotency and early differentiation, as the biological significance of the cell types used here are limited, as described above. The website would benefit from a user guide and detailed descriptions or figure legends so that the graphs that are generated can be interpreted appropriately.

Reply: We agree and modified the interface so that it can analyze datasets provided by other researchers on the condition that these datasets would be uploaded with the same structure as ours. The new online version now includes an option to select either our or to upload own dataset. The selection of transitions is done in the same way as when using our own data.

Additionally, to guide users through submitting their own data and interpreting the results displayed by the interface, we provided a new Extended Data Figure 1. The figure describes the structure of the datasets that needs to be followed for submission. We simplified the data format as much as possible so that the interface can be widely used. Three standard columns comprising protein ID, Gene names and unique peptides number need to be specified which is a standard in the field. Then the data must include two categories starting with Exp_ and Sm_ identifiers followed by the condition (“cell type” in our case) and replicate number. The Exp_ and Sm_ identifiers refer to protein expression and stability that we studied in the article however, any two quantitative parameters can be studied with the interface instead, as long as the Sm_ and Exp_ identifiers are kept as they are. The condition, “cell type” in our case, can be replaced by any other condition name (drug treatment, disease...). Any number of replicates and conditions can be uploaded given that the name of the condition is kept.

In Extended Data Figure 1b, we provide a step-by-step user guide for the elements displayed on the interface. We describe the content of each plot and the options that can be selected, the method to upload the data and choices for the comparisons. We also provide a detailed explanation for the interpretation of each graph and the options that are available to the user. This figure was also added to the interface with its caption as suggested by reviewer 3 (see Extended Data Fig. S1a).

3.2.2. Plots that can be downloaded as SVG would be more useful than PDF.

Reply: We also added the option to export graphs in SVG format.

3.3-In the manuscript figures, bar graphs are not appropriate for displaying quantitative data. Rather, dot blots or similar format which displays datapoints should be used.

Reply: We agree and modified the figures accordingly. (same comment as 1.7, 1.8 and 1.15)

References:

1. Becher, I. *et al.* Pervasive Protein Thermal Stability Variation during the Cell Cycle. *Cell* **173**, 1495-1507.e18 (2018).
2. Dai, L. *et al.* Modulation of Protein-Interaction States through the Cell Cycle. *Cell* **173**, 1481-1494.e13 (2018).
3. T, E. & P, V. Polysome translational state during the cell cycle. *Eur. J. Biochem.* **52**, 203–210 (1975).
4. Haneke, K. *et al.* CDK1 couples proliferation with protein synthesis. *J. Cell Biol.* **219**, (2020).
5. Calamita, P. *et al.* SBDS-Deficient Cells Have an Altered Homeostatic Equilibrium due to Translational Inefficiency Which Explains their Reduced Fitness and Provides a Logical Framework for Intervention. *PLoS Genet.* **13**, e1006552 (2017).
6. Tulpule, A. *et al.* Pluripotent Stem Cell Models of Shwachman-Diamond Syndrome Reveal a Common Mechanism for Pancreatic and Hematopoietic Dysfunction. *Cell Stem Cell* **12**, 727–736 (2013).
7. Finch, A. J. *et al.* Uncoupling of GTP hydrolysis from eIF6 release on the ribosome causes shwachman-diamond syndrome. *Genes Dev.* **25**, 917–929 (2011).
8. Sezgin, G. *et al.* Impaired growth, hematopoietic colony formation, and ribosome maturation in human cells depleted of Shwachman-Diamond syndrome protein SBDS. *Pediatr. Blood Cancer* **60**, 281–286 (2013).
9. Brickman, J. M. & Serup, P. Properties of embryoid bodies. *Wiley Interdiscip. Rev. Dev. Biol.* **6**, 259 (2017).
10. Kim, I. S. *et al.* Parallel Single-Cell RNA-Seq and Genetic Recording Reveals Lineage Decisions in Developing Embryoid Bodies. *Cell Rep.* **33**, 108222 (2020).
11. van Wilgenburg, B., Browne, C., Vowles, J. & Cowley, S. A. Efficient, Long Term Production of Monocyte-Derived Macrophages from Human

- Pluripotent Stem Cells under Partly-Defined and Fully-Defined Conditions. *PLoS One* **8**, 71098 (2013).
12. Moreau, T. *et al.* Large-scale production of megakaryocytes from human pluripotent stem cells by chemically defined forward programming. *Nat. Commun.* **7**, 1–16 (2016).
 13. Tohyama, S. *et al.* Distinct metabolic flow enables large-scale purification of mouse and human pluripotent stem cell-derived cardiomyocytes. *Cell Stem Cell* **12**, 127–137 (2013).
 14. Guadix, J. A. *et al.* Human Pluripotent Stem Cell Differentiation into Functional Epicardial Progenitor Cells. *Stem Cell Reports* **9**, 1754–1764 (2017).
 15. Lian, X. *et al.* Robust cardiomyocyte differentiation from human pluripotent stem cells via temporal modulation of canonical Wnt signaling. *Proc. Natl. Acad. Sci. U. S. A.* **109**, E1848–E1857 (2012).
 16. Koch, P., Opitz, T., Steinbeck, J. A., Ladewig, J. & Brüstle, O. A rosette-type, self-renewing human ES cell-derived neural stem cell with potential for in vitro instruction and synaptic integration. *Proc. Natl. Acad. Sci. U. S. A.* **106**, 3225–3230 (2009).
 17. Sugimura, R. *et al.* Haematopoietic stem and progenitor cells from human pluripotent stem cells. *Nature* **545**, 432–438 (2017).
 18. Lachmann, N. *et al.* Large-scale hematopoietic differentiation of human induced pluripotent stem cells provides granulocytes or macrophages for cell replacement therapies. *Stem Cell Reports* **4**, 282–296 (2015).
 19. Shoichet, B. K., Baase, W. A., Kuroki, R. & Matthews, B. W. A relationship between protein stability and protein function. *Proc. Natl. Acad. Sci.* **92**, 452–456 (1995).
 20. Petukh, M., Kucukkal, T. G. & Alexov, E. On Human Disease-Causing Amino Acid Variants: Statistical Study of Sequence and Structural Patterns. *Hum. Mutat.* **36**, 524–534 (2015).
 21. Kucukkal, T. G., Petukh, M., Li, L. & Alexov, E. Structural and physico-chemical effects of disease and non-disease nsSNPs on proteins. *Curr. Opin. Struct. Biol.* **32**, 18–24 (2015).
 22. Jarzab, A. *et al.* Meltome atlas—thermal proteome stability across the tree of life. *Nat. Methods* **17**, 495–503 (2020).

REVIEWERS' COMMENTS

Reviewer #1 (Remarks to the Author):

The authors have addressed most of my comments. However, a few of them still need to be fixed.

1. The authors replied that “Reply: We agree. The text is corrected to “There are doubts that it was caused by a difference in PTMs, as the latter would reduce the level of unmodified peptides, which may result in altered expression read-outs”.”. The text in the manuscript, however, remains unchanged (page 15, line 330).

2. The authors added Fig. S4b to show the relationship between the fold changes in stability vs expression. The conclusion is “there was no correlation between the FCs in stability versus expression for individual proteins”. A precise conclusion should be put down this way “we did not observe a significant correlation between the FCs in stability versus expression for individual proteins.” It would require certain statistical analysis to conclude “there was no correlation....”

3. Table S3, the authors provided the QC pdf files from TPP package. But these files are dataset level QC, and don't show the quality of the fitted melting curves for each protein. Statistics, such as R², SSE, plateau, should be appended to Table S3 for each protein. These statistics are in the output table from TPP package.

4. Fig. 2e, regarding to the comment on the loose p value cutoff, the authors said “A relatively loose cut-off in Fig. 2e was employed on purpose, as the selected proteins passed through the additional filter, Gene Ontology classification. The sufficiency of the chosen statistical threshold is supported by the high enrichment of the GO terms for many transitions: e.g., for B->A, the enrichment was 20-40 fold. In addition, the Fischer score cutoff can be adjusted manually on the web interface to fit the user's preferences.”. “Gene Ontology classification” is not a filter. It is simply a way of annotating genes. Besides, among the 22 highlighted transitions, B->A is the only one that reaches 20-40 fold. Most of them are 0-5 fold. This is not a proper justification for the loose p value cutoff. Moreover, the p values are not corrected for multiple hypothesis testing.

5. Please rasterize the Sankey diagrams and the scatter plots. It is super hard to view some of the figures. It takes forever for an average computer to display them.

Reviewer #2 (Remarks to the Author):

The authors have not addressed my comments by performing additional experiments, but rather chose to refer to published papers. Notwithstanding this weakness, I think they generated a large amount of solid data that merit publication in Nature Communications.

Reviewer #3 (Remarks to the Author):

My concerns have been adequately addressed. The messaging of the manuscript has been improved. The online tool should be of interest to a broad audience.

REVIEWERS' COMMENTS

Reviewer #1 (Remarks to the Author):

The authors have addressed most of my comments. However, a few of them still need to be fixed.

1. The authors replied that "Reply: We agree. The text is corrected to "There are doubts that it was caused by a difference in PTMs, as the latter would reduce the level of unmodified peptides, which may result in altered expression read-outs"". The text in the manuscript, however, remains unchanged (page 15, line 330).

Reply: We apologise for this miss; we made sure the sentence is now added (page 14 line 10).

2. The authors added Fig. S4b to show the relationship between the fold changes in stability vs expression. The conclusion is "there was no correlation between the FCs in stability versus expression for individual proteins". A precise conclusion should be put down this way "we did not observe a significant correlation between the FCs in stability versus expression for individual proteins." It would require certain statistical analysis to conclude "there was no correlation...."

Reply: We agree with this comment and corrected the sentence.

3. Table S3, the authors provided the QC pdf files from TPP package. But these files are dataset level QC, and don't show the quality of the fitted melting curves for each protein. Statistics, such as R², SSE, plateau, should be appended to Table S3 for each protein. These statistics are in the output table from TPP package.

Reply: We added the TPP package output table to supplementary Table S3.

4. Fig. 2e, regarding to the comment on the loose p value cutoff, the authors said "A relatively loose cut-off in Fig. 2e was employed on purpose, as the selected proteins passed through the additional filter, Gene Ontology classification. The sufficiency of the chosen statistical threshold is supported by the high enrichment of the GO terms for many transitions: e.g., for B->A, the enrichment was 20-40 fold. In addition, the Fischer score cutoff can be adjusted manually on the web interface to fit the user's preferences.". "Gene Ontology classification" is not a filter. It is simply a way of annotating genes. Besides, among the 22 highlighted transitions, B->A is the only one that reaches 20-40 fold. Most of them are 0-5 fold. This is not a proper justification for the loose p value cutoff. Moreover, the p values are not corrected for multiple hypothesis testing.

Reply: We apologize for the misunderstanding and thank reviewer 1 for pointing this out. For classifying proteins into sectors, we used a standard p-value threshold of 0.05 (widely used in biology) after *t*-tests for stability and expression were combined by Fischer's formula (an established method to combine p-value of independent measurements)

(Fisher, R.A. (1925). Statistical Methods for Research Workers. Oliver and Boyd (Edinburgh). ISBN 0-05-002170-2) (also available here¹). In addition, to evaluate the false positive rates when such a cut-off is applied, a permutation analysis of the replicates was performed, such as the one suggested previously by Reviewer 1 to justify our cut-off in Fig. 7f. Using the 0.05 cut-off for transitions from A-D to A-D between iPSCs, EBs and hFF (same sample groups as in Fig. 2e), we found 96 significant proteins out of 46 668 unique permutations (6 x 7778 proteins). This could be compared to 2589 significant proteins out of 7778 in Fig. 2e, when replicates are allocated to the right sample groups. This result confirms that the chosen cut-off provides low rate of false positives (<1% in this case). Lastly, on the web interface producing the Sankey diagram, the limit for the lowest p-value was modified so that the user can further increase the stringency of the analysis. Therefore, if the user is not satisfied with the standard 5% cut-off, he can easily increase the threshold for significant hits and thus lower the p-values of the hits down to the lowest value of 0.001. This new feature is specified in Supplementary Fig. 1.

5. Please rasterize the Sankey diagrams and the scatter plots. It is super hard to view some of the figures. It takes forever for an average computer to display them.

Reply: We agree and rasterized all the figures.

Reviewer #2 (Remarks to the Author):

The authors have not addressed my comments by performing additional experiments, but rather chose to refer to published papers. Notwithstanding this weakness, I think they generated a large amount of solid data that merit publication in Nature Communications.

Reply: We thank Reviewer 2 for the kind comments.

Reviewer #3 (Remarks to the Author):

My concerns have been adequately addressed. The messaging of the manuscript has been improved. The online tool should be of interest to a broad audience.

Reply: We thank Reviewer 3 for the kind comments.

References

1. Fisher, R. A. Statistical Methods for Research Workers. in 66–70 (Springer, New York, NY, 1992). doi:10.1007/978-1-4612-4380-9_6.